

# Page curve for entanglement negativity
# through geometric evaporation

**Jaydeep Kumar Basak[1], Debarshi Basu[1], Vinay Malvimat[2⋆],**
**Himanshu Parihar[1] and Gautam Sengupta [1]**

**1** Department of Physics, Indian Institute of Technology Kanpur, 208016, India
**2** Indian Institute of Science Education and Research, Homi Bhabha Rd,
Pashan, Pune 411 008, India

⋆ vinaymmp@gmail.com

## Abstract

We compute the entanglement negativity for various pure and mixed state configurations in a bath coupled to an evaporating two dimensional non-extremal Jackiw-Teitelboim (JT) black hole obtained through the partial dimensional reduction of a three dimensional BTZ black hole. Our results exactly reproduce the analogues of the Page curve for the entanglement negativity which were recently determined through diagrammatic technique developed in the context of random matrix theory.

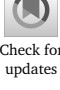

## Contents

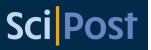

# 1   Introduction

A clear resolution to the black hole information paradox is essential to understanding several intriguing aspects of semiclassical and quantum gravity. A recent progress towards a possible solution to the puzzle involves the appearance of certain regions known as *"islands"* in the black hole space time which contribute significantly to the fine grained entropy of the Hawking radiation at late times, leading to a unitary Page curve [1]. The island proposal was inspired by earlier works on the quantum correction to the Ryu-Takayanagi proposal for the holographic entanglement entropy computed through a *quantum extremal surface* which is obtained by extremizing the *generalized entropy* given by the sum of the area of a codimension 2 surface homologous to the subsystem under consideration and the bulk entanglement entropy across that surface as described in [2–4]. The island formulation has led to many exciting developments ranging from the role of the space-time replica wormholes in the computation of the Euclidean gravitational path integrals [5, 6] to the reproduction of the Page curve. Obtaining the Page curve indicates towards an unitary evolution during the black hole formation and evaporation process, and hence at the possibility of a resolution to the information loss paradox ( The literature in this exciting area is vast. See [7, 8] and the references therein). The above discussed island formula for computing the fine grained/entanglement entropy of a subregion in a $d$ dimensional quantum field theory coupled to semiclassical gravity is given by

$$S[\mathrm{X}] = \min \left\{ \mathrm{ext}_{Is(X)} \left[ \frac{\mathrm{Area}[\partial Is(X)]}{4 G_N} + S_{eff}[\mathrm{X} \cup Is(X)] \right] \right\}, \tag{1}$$

where-$X$ is the subregion under consideration, $Is(X)$ is the island corresponding to given region-$X$, $G_N$ denotes the Newton's gravitational constant and $S_{eff}(Y)$ corresponds to the effective semiclassical entanglement entropy of quantum matter fields in $Y$.

    The island formulation can be understood very naturally in the frame work of *double holography*. As discussed in [1], in the double holographic scenario, the $d$ dimensional quantum field theory coupled to semiclassical gravity, is itself holographically dual to a $d + 1$ dimensional gravitational theory. In this context, the entanglement entropy determined utilizing the above mentioned island formula in the $d$-dimensional theory, is simply given by the RT/HRT formula in the dual $d + 1$ dimensional bulk space time. In fact, as argued in [1], the application of the RT/HRT formula in the $d + 1$ dimensional bulk space time clearly suggests that the island present in the black hole interior is a part of the entanglement wedge of the Hawking radiation/bath which leads to a realization of the ER=EPR scenario [9]. This nice feature of double holography naturally leads to a geometric perspective of the black hole formation and evaporation. Since this exciting development, a variety of very interesting models have

explored different techniques to geometrize the island formulation [10–25]. In this article we focus on another distinct recent geometric construction proposed in [20]. The authors in [20] motivated by [26], demonstrated that the evaporation of a two dimensional Jackiw-Teitelboim black hole could be modelled by a partial dimensional reduction of a three dimensional BTZ black hole. In this model the extent to which the JT black hole evaporates is controlled by the parameter describing the partial dimensional reduction.

We know from quantum information theory that the above mentioned entanglement entropy is an unique entanglement measure for pure quantum states only. However, for mixed quantum states, various quantum information theoretic measures characterizing distinct quantum/classical properties have been proposed [27, 28]. Remarkably a good number of these measures such as the entanglement of purification, odd entropy, reflected entropy admit holographic descriptions as suggested in [29–31]. Recently, island formulations have been proposed for each of these measures to further investigate and explore the mixed state entanglement and correlation structure present in the Hawking radiation [12, 32, 33]. Another significant computable mixed state entanglement measure in this regard is the *entanglement negativity* [34]. This particular measure is a non-convex entanglement monotone which characterizes the upper bound on the distillable entanglement of the given quantum state [34,35]. A replica technique was then developed to obtain this quantity in quantum field theories utilizing which the explicit computation was performed for various configurations in two dimensional conformal field theories [36–38]. Furthermore, this led to the first attempt for the holographic entanglement negativity of pure states made in [39]. Soon after another distinct holographic conjecture involving the algebraic sum of the areas of the extremal surfaces for various pure and mixed state configurations of the adjacent, the disjoint and the single intervals in a zero and finite temperature $CFT_{1+1}$ was proposed in a series of articles [40–47]. Recently, this proposal was re-expressed in terms of the areas of the combination of backreacting cosmic branes corresponding to Renyi entropy of order half [48]. Apart from the above mentioned proposals another alternative holographic construction for the entanglement negativity involving the backreacting entanglement wedge cross section (EWCS) or the Reflected entropy of order half was developed in [49,50]. Note that the results from the holographic proposals involving the algebraic sum of the areas of the extremal surfaces [40,43,45], precisely match with the corresponding results obtained using the EWCS in [49–52]. In a recent article [53], the authors computed the gravitational path integral corresponding to the Renyi negativities in holographic theories and demonstrated that it is dominated by a bulk replica symmetry breaking saddle. Following this in [48], the authors of the present article demonstrated that for two dimensional holographic $CFTs$ and for the case of subsystems involving spherical entangling surfaces in higher dimensions, the computation in [53] also serves as a proof of the holographic proposals [40,43,45].

Inspired by the above mentioned holographic constructions, two alternative island formulations for the entanglement negativity, first one involving the extremization of an algebraic sum of the generalized Renyi entropy of order half and the second one involving the sum of the backreacting EWCS and the effective entanglement negativity, were proposed by the authors of the present article in [48]. We demonstrated that the expressions for the entanglement negativity obtained by the two proposals for various pure and mixed state configurations in baths coupled to extremal and non-extremal JT black holes, match exactly and also with the generalized Renyi reflected entropy of order half as expected. Furthermore, we provided a possible derivation of our island proposal for the entanglement negativity involving the areas of backreacting cosmic branes by considering the replica wormhole contributions to the gravitational path integral involving the replica symmetry breaking saddle of [53] .

The connection between random matrix theory and black holes has gained utmost significance in recent times in several exciting phenomena explored in [5, 54–62]. Quite inter-

estingly, several such techniques involving Harr random unitaries have been utilized to gain insight into the black hole evaporation and the information loss paradox (see [63–69] and references therein). More recently, very interesting developments have led to the computation of various measures such as the entanglement negativity, mutual information, relative entropy to obtain analogues of the Page curve for these measures in Harr random states [70,71]. In [70] the authors have developed a systematic diagrammatic technique to obtain the entanglement negativity of a a bipartite system in a random mixed state chosen from the Wishart ensemble. This led them to determine the analogues of Page curves for the entanglement negativity for bipartite systems in random mixed states. Furthermore, the authors provided an interpretation for the specific behaviours of the entanglement negativity in terms of the dimensions of the Hilbert spaces of the subsystems involved. The analogues of such Page curves for the entanglement negativity thus obtained for various configurations of the random states is expected to shed new light on the structure of mixed state entanglement in Hawking radiation. Hence, the discussion above raises the crucial issue of obtaining the analogues of the Page curve for the entanglement negativity in the context of black hole evaporation and compare them to the corresponding results obtained through random matrix technique in [70].

In this article we address this significant issue by computing the entanglement negativity of various mixed state configurations in a bath coupled to a two dimensional non-extremal JT black hole obtained by a partial dimensional reduction of a three dimensional BTZ black hole as described in [20]. We will demonstrate that the entanglement negativity of various bipartite systems in a bath coupled to a non-extremal JT black hole exactly reproduces the analogues of the Page curve for the entanglement negativity of random mixed states which were obtained in [70]. Furthermore we interpret the results we derived in terms of the Hawking quanta collected in various subsystems in the bath and their respective islands.

The article is organized as follows. In section 2 we review the model involving the partial dimensional reduction described in [20] to obtain the Page curve for the entanglement entropy of the entire bath/radiation coupled to a JT black hole. Following this we briefly review [70] where the authors obtain the entanglement negativity of a random bipartite mixed state and then go on to describe the holographic entanglement negativity construction proposed in [43, 45]. In section 3 we describe our computation of the entanglement negativity of the pure and mixed state configurations involving disjoint, adjacent and single intervals in a bath $CFT_2$ coupled to an evaporating non-extremal JT gravity obtained through the partial dimensional reduction of the BTZ black hole. We will demonstrate that the result we obtain exactly reproduces the analogue of the Page curves for entanglement negativity of the random mixed states obtained in [70]. In section 4 we summarize the results we have obtained and discuss the plausible future directions. Furthermore, in appendix A we determine the entanglement negativity for various pure and mixed state configurations in a bath $CFT_2$ coupled to an extremal JT black hole obtained through the partial dimensional reduction of the pure $AdS_3$ spacetime. However, a clear interpretation of the results we obtain for configurations in a bath coupled to an extremal JT black hole, remains an open issue for future investigations.

## 2   Review of the Earlier Results

In this section, we review the computation of the entanglement entropy of a bath coupled to an evaporating two dimensional black hole in JT gravity obtained through a partial dimensional reduction of the three dimensional BTZ black hole in [20]. The authors in [20] introduce the time dependence in the two dimensions by varying the parameter describing the dimensional reduction. We will explain below the detail of their construction which reproduces the Page curve for the entanglement entropy and hence, provides a geometric representation of

the black hole evaporation process. We then briefly describe the entanglement negativity of a bipartite system in a random mixed state and the corresponding Page curve derived using the random matrix techniques in [70]. Subsequently, we discuss the holographic entanglement negativity proposal for various mixed state configurations in the AdS$_3$/CFT$_2$ scenario conjectured in [40, 43, 45].

## 2.1 Page Curve for Entanglement Entropy through Geometric Evaporation

### 2.1.1 Non-Extremal JT black hole from BTZ

The authors in [20], considered the BTZ black hole whose metric is given by

$$ds^2 = -\left(\frac{r^2 - r_h^2}{L_3^2}\right) dt^2 + \left(\frac{r^2 - r_h^2}{L_3^2}\right)^{-1} dr^2 + r^2 d\varphi^2,  \tag{2}$$

where $r_h$ is the horizon radius, $L_3$ denotes the $AdS_3$ length scale and the angular direction $\varphi$ is periodic i.e $\varphi \sim \varphi + 2\pi$. The length of a geodesic homologous to a boundary subsystem described by an angular interval of extent $\Delta\varphi$ is given by

$$\mathcal{L}_{\Delta\varphi, Con1} = 2L_3 \log\left[\sinh\frac{r_h \Delta\varphi}{2L_3}\right] + \text{UV cutoff},  \tag{3}$$

$$\mathcal{L}_{\Delta\varphi, Con2} = 2L_3 \log\left[\sinh\frac{r_h(2\pi - \Delta\varphi)}{2L_3}\right] + \text{UV cutoff},  \tag{4}$$

where $\Delta\varphi$ is the angle subtended by the subsystem under consideration and $Con1$ denotes a connected RT surface which corresponds to a geodesic. As described in [20, 72, 73], there is another extremal surface which also satisfies the homology condition. This is a disconnected extremal surface which involves the horizon of the BTZ black hole and the geodesic anchored to the complement of the subsystem considered. Note that the authors considered the BTZ black hole to be formed from collapsing matter which was in a pure quantum state. Hence, they did not include the contribution from the black hole horizon in the second extremal surface as it would correspond to a dual CFT characterized by a mixed quantum state. The resulting extremal surface after discarding the contribution from black hole horizon therefore, is a connected geodesic which is denoted by the suffix $Con2$ in eq. (4). The entanglement entropy is then given by the minimum of these two geodesic lengths as expressed below

$$S_{\Delta\varphi} = \frac{1}{4G_N^{(3)}} Min\left[\mathcal{L}_{\Delta\varphi, Con1}, \mathcal{L}_{\Delta\varphi, Con2}\right].  \tag{5}$$

The BTZ metric in eq. (2) corresponds to an asymptotically $AdS$ space time whose action is given by

$$S = \frac{1}{16\pi G^{(3)}} \int d^3x \sqrt{-g}\left(R^{(3)} - 2\Lambda\right),  \tag{6}$$

where the cosmological constant $\Lambda < 0$ and $G^{(3)}$ corresponds to the three dimensional Newton's gravitational constant. Since, the BTZ metric in eq. (2) does not depend on the $\varphi$ coordinate it could be expressed in the following form

$$ds^2 = g_{\mu\nu} dx^\mu dx^\nu = h_{ab}(x^a) dx^b dx^b + \phi^2(x^a) L_3^2 d\varphi^2,  \tag{7}$$

where $\mu, \nu = 0, 1, 2$ and $a, b = 0, 1$. In [26], it was demonstrated that a dimensional reduction may be performed because the BTZ black hole metric may be re-expressed in the above form.

This led the authors in [20] to integrate the angular direction $\phi$ in the action given by eq. (6) to obtain

$$S = \frac{2\pi\alpha L_3}{16\pi G^{(3)}} \int d^2x \sqrt{-h} \phi \left(R^{(2)} - 2\Lambda\right), \tag{8}$$

In the above equation $\alpha \in (0, 1]$ denotes the parameter which controls the extent of partial dimensional reduction. The metric after the reduction may be expressed as

$$ds^2 = -\frac{4\pi^2 L^2}{\beta^2} \frac{du\,dv}{\sinh^2 \frac{\pi}{\beta}(u-v)} + \frac{4\pi^2 L^4}{\beta^2} \frac{1}{\tanh^2 \frac{\pi}{\beta}(u-v)} d\varphi^2. \tag{9}$$

Observe that the first term in the above expression corresponds to the metric of the non-extremal black hole in JT gravity upon identifying $L_3$ with $L$ which is the $AdS_2$ length scale. The two dimensional black hole in JT gravity inherits the temperature from the BTZ black hole upon dimensional reduction. After integrating out $\varphi$ as described by eq. (8), the dilaton profile is given by

$$\Phi = \Phi_0 + \frac{2\pi\Phi_r}{\beta} \coth \frac{\pi}{\beta}(u-v), \tag{10}$$

where $u = t + r^*$ and $v = t - r^*$ are the light-cone coordinates, with $r^*$ being the usual radial tortoise coordinate [20]. In eq. (10), $\Phi_r = 2\pi L\alpha$ is the renormalized value of the dilaton which carries the signature of the dimensional reduction through the parameter $\alpha$. Note that in order to arrive at the above expression, the authors utilized the relations $L_3 = L$, $G^{(3)} = LG^{(2)}, r_h = \frac{2\pi L^2}{\beta}$ in eq. (3) and eq. (4), and absorbed the UV cut off in the background value of the dilaton field $\Phi_0$.

The authors then utilized the expressions for geodesic lengths described in equations (3) and (4) and then incorporated the appropriate substitutions for partial dimensional reduction. Upon performing the partial reduction, the interval $[0, b]$ correspond to the quantum mechanical degrees of freedom dual to the JT black hole in the limit $b \to 0$ and the rest of the system corresponds to that of the bath/radiation. This led the authors to obtain the entanglement entropy of the entire radiation/bath ($S_{Rad}$) to be as follows

$$S_{Rad,\ Con1} = \frac{1}{4G_N^{(2)}} \left(\Phi_0 + 2\log\left[\sinh\frac{\pi}{\beta}(2\pi L(1-\alpha) - 2b)\right]\right),$$

$$S_{Rad,\ Con2} = \frac{1}{4G_N^{(2)}} \left(\Phi_0 + 2\log\left[\sinh\frac{\pi}{\beta}(2\pi L\alpha + 2b)\right]\right),$$

$$S_{Rad} = Min\left[S_{Rad,\ Con1},\ S_{Rad,\ Con2}\right], \tag{11}$$

As explained earlier the subscripts $Con1$ and $Con2$ denote the two possible connected RT surfaces. Note that the entanglement entropy above is expressed in terms of the dimensional reduction parameter $\alpha$. The authors in [20] developed a dynamical evaporation scheme for the JT black hole by putting time dependence on the parameter $\alpha$. Varying $\alpha$ is tantamount to a time-dependent renormalized dilaton[1]

$$\Phi_r(t) = 2\pi L\alpha(t) = 2\pi L\left(1 - \frac{A}{2}t\right), \tag{12}$$

---

[1]Note that the coordinate in which the dilaton becomes time-dependent was denoted by $\tilde{t}$ in [20]. In the present article we have chosen to omit the tildes for brevity.

where $A = \frac{c}{6}\frac{G_N}{\pi L}$ and $c$ is the central charge of the matter CFT$_2$ which is obtained as the holographic dual of the rest of the BTZ black hole spacetime that has not been dimensionally reduced. The central charge of a holographic $CFT_2$ is related to the three dimensional gravitational constant $G_N$ through the well known Brown Henneaux formula $c = \frac{3L}{2G_N}$ [74]. As shown in [20], the conformal anomaly present in the matter CFT$_2$ gives rise to an outgoing energy flux which simulates a black hole evaporation process within the JT gravity framework. Subsequently, the authors utilized their construction to demonstrate that the entanglement entropy of the bath/radiation given by eq. (11), follows the Page curve during the evaporation of the non extremal black hole in JT gravity.

## 2.2 Page Curve for Entanglement Negativity from Random Matrix Theory

In this subsection we present a concise review of the entanglement properties of random mixed states as described in [70]. The authors developed a diagrammatic technique within the framework of random matrix theory and investigated the entanglement negativity of a random bipartite mixed state. The basic setup in [70] consists of a tripartite system which we denote as $R_1 \cup R_2 \cup B$, and in the context of present article, we interpret $B$ as a black hole while $R \equiv R_1 \cup R_2$ constitutes the bath [2]. The random mixed nature of $R$ was produced by varying the size of the subsystems involved. It was shown in [70] that for large Hilbert space dimensions, the random reduced density matrix of the subsystem $R$ which in the context of present article represents the radiation/bath is captured by the Wishart ensemble [75]:

$$\rho_R \approx XX^\dagger, \tag{13}$$

where $X$ is a (random) $dim\,\mathcal{H}_R \times dim\,\mathcal{H}_B$ rectangular matrix sampled from a Gaussian distribution. Subsequently, utilizing a diagrammatic implementation of the partial transposition, various moments of the density matrix $\rho_R^{T_2}$ were computed in the limit of large Hilbert space dimensions. The ensemble averaged entanglement negativity of the mixed state described by $R$ may then be expressed in terms of the number of qubits, $N$ in the respective systems as

$$\left\langle \mathcal{E}_{R_1 R_2} \right\rangle \approx \begin{cases} \frac{1}{2}(N_R - N_B), & N_{R_{1,2}} < \frac{N}{2} \\ \min\,(N_R\,,\,N_B), & \text{otherwise} \end{cases}, \tag{14}$$

for $N_R > N_B$, while for $N_R < N_B$ it turns out to be vanishingly small. The authors of [70] obtained several Page-like curves by varying the size of $R_1$ and $R_2$ [3] relative to $B$. In the present work we will exactly reproduce the above discussed analogues of the Page curves for entanglement negativity, but in the context of an evaporating non-extremal JT black hole coupled to a bath, which is obtained by a partial dimensional reduction of a three dimensional BTZ black hole.

## 2.3 Holographic Entanglement Negativity

In this subsection we briefly recall the salient features of the holographic entanglement negativity proposals in [40,43,45]. Based on the replica technique computations of the entanglement negativity in 2d CFTs [36,37], it was argued that the holographic entanglement negativity (HEN) for various bipartite mixed states was given by an algebraic sum of the lengths of a

---

[2]Note that originally in [70] the bipartite system was denoted as $AB$, where $B$ was interpreted as the auxiliary bath subsystem.

[3]Here changing the lengths of subsystems essentially correspond to changing the dimensions of the respective Hilbert spaces.

specific combination of bulk geodesics homologous to different sub-regions in the dual $CFT_2$. For example, the HEN for two adjacent intervals $A$ and $B$ was given by [43, 48]

$$\mathcal{E}(A:B) = \frac{3}{16\pi G_N^{(3)}}(\mathcal{L}_A + \mathcal{L}_B - \mathcal{L}_{A\cup B}) \tag{15}$$

$$= \frac{3}{4}[S(A) + S(B) - S(A\cup B)], \tag{16}$$

where $\mathcal{L}_X$ is the length of the minimal surface (geodesic) homologous to subsystem $X$, and in the second step we have made use of the Ryu-Takayanagi formula [76, 77]. As the entanglement negativity of a pure state is given by the Renyi entropy of order half, we wish to re-express eq. (15) in terms of the Renyi entropies of order half. To this end, we recall that the Renyi entropy in a holographic $CFT_d$ is dual to the area of a backreacting cosmic brane $\mathcal{A}_A^{(n)}$ [4], in the dual bulk $AdS_{d+1}$ spacetime, homologous to the subsystem-$A$ in question [78]

$$S^{(n)}(A) = \frac{\mathcal{A}_A^{(n)}}{4G_N^{(d+1)}}, \tag{17}$$

where $G_N^{(d+1)}$ is the $(d+1)$ dimensional Newton's constant. Hence, in the context of $AdS_3$/$CFT_2$, the Renyi entropy of order half is given by

$$S^{(1/2)}(A) = \frac{\mathcal{L}_A^{(1/2)}}{4G_N^{(3)}}, \tag{18}$$

where $\mathcal{L}_A^{(1/2)}$ is related to the length of a back reacting cosmic brane in $AdS_3$ space time, which is homologous to an interval-$A$. As described in [39, 49, 79] for spherical entangling surfaces and for a subsystem in a holographic $CFT_2$, the effect of the backreaction can be quantified specifically

$$\mathcal{L}_A^{(1/2)} = \frac{3}{2}\mathcal{L}_A. \tag{19}$$

Now, we may utilize eqs. (18) and (19) to rewrite eq. (15) for the holographic entanglement negativity between two adjacent intervals in the following instructive form

$$\mathcal{E}(A:B) = \frac{1}{2}\left[S^{(1/2)}(A) + S^{(1/2)}(B) - S^{(1/2)}(A\cup B)\right]. \tag{20}$$

In a similar manner we can re-express the holographic conjecture in [45] for the entanglement negativity of the mixed state configuration of two disjoint intervals $A$ and $B$ in proximity[5], in a holographic $CFT_2$, as

$$\mathcal{E} = \frac{1}{8G_N}\left[\mathcal{L}_{A\cup C}^{(1/2)} + \mathcal{L}_{B\cup C}^{(1/2)} - \mathcal{L}_{A\cup B\cup C}^{(1/2)} - \mathcal{L}_C^{(1/2)}\right] \tag{21}$$

$$= \frac{1}{2}\left[S^{(1/2)}(A\cup C) + S^{(1/2)}(B\cup C) - S^{(1/2)}(A\cup B\cup C) - S^{(1/2)}(C)\right]. \tag{22}$$

The subsystem $C$ in the above equation denotes an interval sandwiched between the two intervals $A$ and $B$. In the following we will utilize these expressions to compute the holographic entanglement negativity for various mixed state configurations involving different sub-regions of the JT gravity + $CFT_2$ obtained via partial dimensional reductions of various $AdS_3$ geometries.

---

[4]Note that $\mathcal{A}^{(n)}$ is not exactly the area of a backreacting brane but is related to it as $n^2\frac{\partial}{\partial n}(\frac{n-1}{n}\mathcal{A}^{(n)}) = \text{Area}(\text{ cosmic brane }_n)$.

[5]Note that in a holographic $CFT_2$ the proximity approximation corresponds to the $x \to 1$ channel (where $x$ is the cross ratio) of the corresponding four point twist correlator as described in [45,80]. In the $x \to 0$ channel which corresponds to the configurations involving the disjoint intervals which are far apart the entanglement negativity vanishes.

# 3 Page Curve for Entanglement Negativity

In this section we compute the entanglement negativity of various bipartite pure and mixed state configurations in a bath coupled to a non-extremal black hole in JT gravity, through the partial dimensional reduction of a BTZ black hole. For the mixed state configurations involving adjacent and disjoint intervals in a bath subsystem, we demonstrate that the entanglement negativity reproduces the Page curves obtained from the random matrix techniques in [70] which was described in previous section.

The Page curve for a bipartite quantum system with a finite dimensional Hilbert space was obtained in [81]. It was determined by plotting the Harr random average of the entanglement entropy as a function of the Hilbert space size for one of the subsystems in the bipartite system. Only later this was interpreted in the context of black hole evaporation by defining the Page curve as the time evolution of the entanglement entropy of the Hawking radiation [82, 83]. Similarly the Page curve for the entanglement negativity derived through the Random matrix technique in [70] discussed earlier is also expressed as a function of the Hilbert space sizes of subsystems involved. Observe that the $\alpha$ parameter which controls the extent to which the black hole evaporates in [20] also describes an angle. In this section, we determine the analogue of such curves for the entanglement negativity by examining its time evolution by varying the $\alpha$ parameter in some cases whereas in other cases we fix $\alpha$ but vary the sizes of the subsystems involved. We emphasize that it is perfectly valid to examine the behaviour entanglement negativity as a function of the size of the subsystems for a fixed $\alpha$ or vary the $\alpha$ parameter. However in order to compare our results with corresponding results in the random matrix theory we kept the $\alpha$ fixed in some of our computations. More specifically in subsection 3.1.1 we consider the time evolution of the entanglement negativity of a single interval in the bath/radiation system. Subsequently we obtain the time evolution of the entanglement negativity for the configuration of the adjacent intervals in bath in subsection 3.1.2 (a). Furthermore, we also determine the behaviour of the entanglement negativity for the adjacent and disjoint intervals in bath by varying sizes of the subsystems involved, in subsections 3.1.2 (b), 3.1.2 (c) and in subsection 3.1.3 respectively.

Note that the linear characteristics of the Page curve for entanglement entropy obtained through the model in [20], is valid for $\beta << \Phi_r^0$ ( $\Phi_r^0 = 2\pi L$ where $L$ is the AdS radius which we have set to unity ). This is clearly expressed in the line above eq.(4.30) of [20]. Away from this approximation, that is for larger values of $\beta$, the Page curve for entanglement entropy obtained through the partial dimensional reduction deviates from its linear behaviour. This will be true for the holographic entanglement negativity which we compute below as it is given by a linear combination of the Renyi entropies of order half. Hence all our plots are also valid within the approximation $\beta << \Phi_r^0$. Away from this approximation, the plots for entanglement negativity differ from those shown in our article.

## 3.1 Non-Extremal JT black hole

### 3.1.1 Single Interval in Bath

As described earlier the holographic entanglement negativity conjecture we are utilizing is expressed as an algebraic sum of the Renyi entropies of order half of various subsystems for the configuration considered. In this subsection we will describe the holographic computation of the Renyi entropy of order half of a subsystem in a $CFT_2$ dual to a BTZ black hole in $AdS_3$ spacetime and subsequently perform the partial dimensional reduction as described in subsection 2.1.1. We may utilize the relations in eq. (3) and eq. (4) to obtain the lengths of the backreacting cosmic branes which are homologous to a generic subsystem in the bath $CFT_2$ radiation as follows

$$\mathcal{L}_{Con11}^{(1/2)}\left(\ell_{\varphi_1},\ell_{\varphi_2}\right)=\mathcal{L}_{Con1}^{(1/2)}\left(\ell_{\varphi_1}\right)+\mathcal{L}_{Con1}^{(1/2)}\left(\ell_{\varphi_2}\right), \tag{23}$$

$$\mathcal{L}_{Con22}^{(1/2)}\left(\ell_{\varphi_1},\ell_{\varphi_2}\right)=\mathcal{L}_{Con2}^{(1/2)}\left(\ell_{\varphi_1}\right)+\mathcal{L}_{Con2}^{(1/2)}\left(\ell_{\varphi_2}\right), \tag{24}$$

$$\mathcal{L}_{Con12}^{(1/2)}\left(\ell_{\varphi_1},\ell_{\varphi_2}\right)=\mathcal{L}_{Con1}^{(1/2)}\left(\ell_{\varphi_1}\right)+\mathcal{L}_{Con2}^{(1/2)}\left(\ell_{\varphi_2}\right), \tag{25}$$

$$\mathcal{L}_{Dis}^{(1/2)}(\ell_{\Delta\varphi})=6L\log\left[\sinh\frac{\pi\ell_{\Delta\varphi}}{2\beta}\right]+\text{UV cutoff}, \tag{26}$$

where the subscripts $Con11$, $Con12$, $Con22$ denote possible connected RT surfaces and $Dis$ denotes the disconnected RT surface. All of these RT surfaces are depicted in fig. 1. Note that in the above equation $\varphi_i$ ($i=1,2$) are the angles subtended by the endpoints of the subsystem of interest such that $\Delta\varphi=|\varphi_1-\varphi_2|$, and we have written $\ell_{\varphi_i}=L\varphi_i$ and $\ell_{\Delta\varphi}=L\Delta\varphi$. $\mathcal{L}_{Con1}^{(1/2)}\left(\ell_{\varphi_i}\right)$ and $\mathcal{L}_{Con2}^{(1/2)}\left(\ell_{\varphi_i}\right)$ are given by the lengths of the geodesics in the BTZ background as follows

$$\mathcal{L}_{Con1}^{(1/2)}\left(\ell_{\varphi_i}\right)=3L\log\left[\sinh\frac{\pi\ell_{\varphi_i}}{\beta}\right]+\text{UV cutoff}, \tag{27}$$

$$\mathcal{L}_{Con2}^{(1/2)}\left(\ell_{\varphi_i}\right)=3L\log\left[\sinh\frac{\pi(2\pi-\ell_{\varphi_i})}{\beta}\right]+\text{UV cutoff}. \tag{28}$$

Finally the Renyi entropy of a given subsystem is simply obtained by the minimum of the lengths of the backreacting cosmic brane on the above four possible homologous surfaces as given below

$$S^{(1/2)}(\ell_{\Delta\varphi})=\frac{1}{4G_N^{(3)}}\min\left[\mathcal{L}_{Con11}^{(1/2)}\left(\ell_{\varphi_1},\ell_{\varphi_2}\right),\ \mathcal{L}_{Con22}^{(1/2)}\left(\ell_{\varphi_1},\ell_{\varphi_2}\right),\ \mathcal{L}_{Con12}^{(1/2)}\left(\ell_{\varphi_1},\ell_{\varphi_2}\right),\ \mathcal{L}_{Dis}^{(1/2)}(\ell_{\Delta\varphi})\right]. \tag{29}$$

Note that in [20], the subsystem under consideration was the entire bath. However, here we will be considering a subregion inside the bath which leads to the four possible backreacting cosmic branes homologous to the subsystem as depicted in fig. 1.

We may now perform the partial dimensional reduction reviewed in the previous section [20] to obtain the Renyi entropy of order half of an arbitrary subsystem $R$ in a bath coupled to JT gravity to be as follows

$$S_{Con11}^{(1/2)}(R)=\frac{1}{4G_N^{(2)}}\left(2\Phi_0+3\log\left[\sinh\left[\frac{\pi}{\beta}(2\pi L\alpha+2b+2\ell_{\phi'})\right]\right]\right.$$
$$\left.+3\log\left[\sinh\left[\frac{\pi}{\beta}(2\pi L\alpha+2b+2\ell_{\phi'}+2\ell_{\Delta\varphi})\right]\right]\right),$$

$$S_{Con22}^{(1/2)}(R)=\frac{1}{4G_N^{(2)}}\left(2\Phi_0+3\log\left[\sinh\left[\frac{\pi}{\beta}(2\pi L(1-\alpha)-2b-2\ell_{\phi'})\right]\right]\right.$$
$$\left.+3\log\left[\sinh\left[\frac{\pi}{\beta}(2\pi L(1-\alpha)-2(b+\ell_{\phi'}+\ell_{\Delta\varphi}))\right]\right]\right),$$

$$S_{Con12}^{(1/2)}(R)=\frac{1}{4G_N^{(3)}}\left(2\Phi_0+3\log\left[\sinh\left[\frac{\pi}{\beta}(2\pi L\alpha+2b+2\ell_{\phi'})\right]\right]\right.$$
$$\left.+3\log\left[\sinh\left[\frac{\pi}{\beta}(2\pi L(1-\alpha)-2(b+\ell_{\phi'}+\ell_{\Delta\varphi}))\right]\right]\right),$$

$$S_{Disc}^{(1/2)}(R)=\frac{1}{4G_N^{(2)}}\left(2\Phi_0+6\log\left[\sinh\left[\frac{\pi\ell_{\Delta\varphi}}{\beta}\right]\right]\right),$$

$$S^{(1/2)}(R)=Min\left[S^{(1/2)}{}_{R,\,Con11},\ S_{R,\,Con22}^{(1/2)},\ S_{R,\,Con12}^{(1/2)},\ S_{R,\,Disc}^{(1/2)}\right], \tag{30}$$

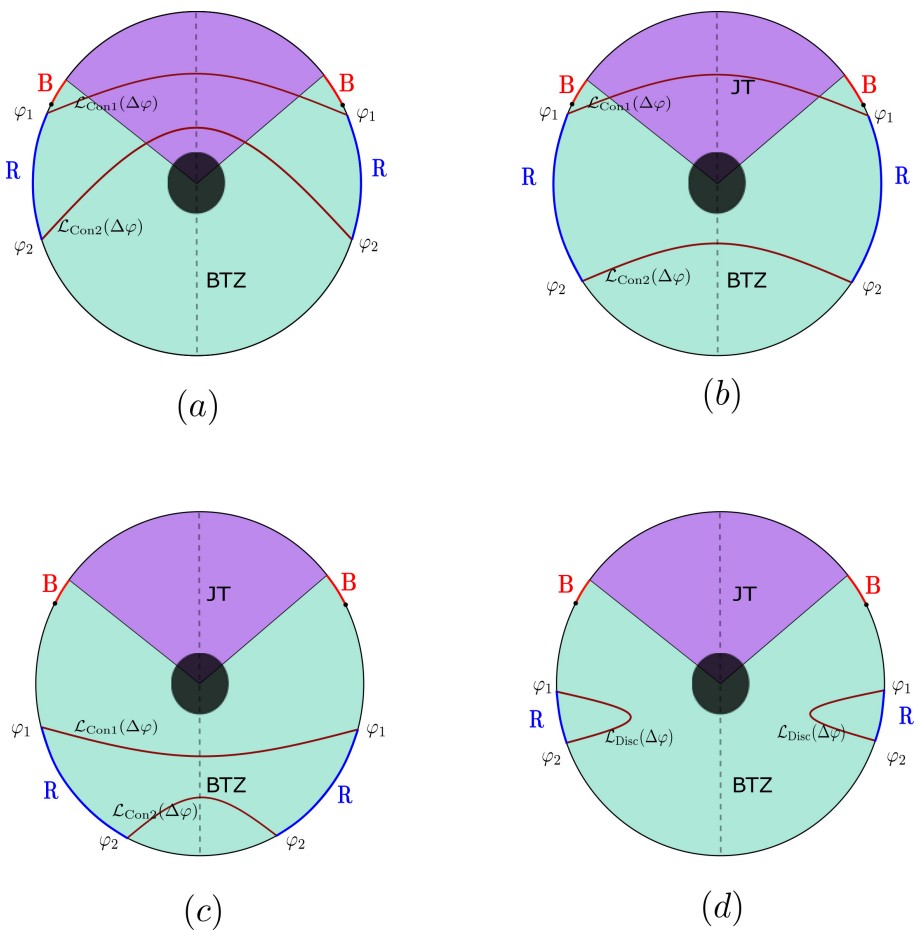

Figure 1: Schematic for possible RT surfaces homologous to the subsystem-$R_i$. Figures (a), (b) and (c) depict the RT surfaces in the connected phase whereas figure (d) corresponds to a schematic of the disconnected phase of the RT surfaces.

where $\ell_{\Delta\varphi} = L\Delta\varphi$ is the length of the interval $R$ and $\ell_{\phi'} = \phi'L$ is its distance from the interval $B = [0, b]$ describing the quantum mechanical degrees of freedom in the limit $b \to 0$ as discussed in the previous section[6].

As described earlier, the authors in [20] considered, the $CFT$ dual to the full three dimensional geometry to be in a pure state. Hence in their model, the entanglement negativity of a single interval with its complement is given by the Renyi entropy of order half as expected from the quantum information theory [36, 37]. Therefore, in the reduced geometry the entanglement negativity of the subsystem $R$ in the bath is given by

$$\mathcal{E}(R) = S^{(1/2)}(R), \tag{31}$$

where $S^{(1/2)}(R)$ is given by eq. (30).

We now compute the entanglement negativity of the subsytem $R$ described by a single interval in the bath depicted in figure 1 by utilizing eq. (31) and eq. (30). The time evolution of the entanglement negativity for this configuration is depicted in the plot given in figure 2, where we have fixed the size of the rest of the bath denoted as $l_2$. As described earlier the entanglement negativity in this case is given by the Renyi entropy of order half and we

---

[6]Note that $[0, b]$ is the interval which correspond to the quantum mechanical degrees of freedom dual to the 2d JT black hole in the limit $b \to 0$. Hence, the length of this interval given by $b$ has to be small. We have introduced $\phi'$ to keep the radiation subsystem chosen to be generic as it need not always be adjacent to the interval $[0, b]$. However, we would like to emphasize that all our results hold even if one sets $\phi' = 0$.

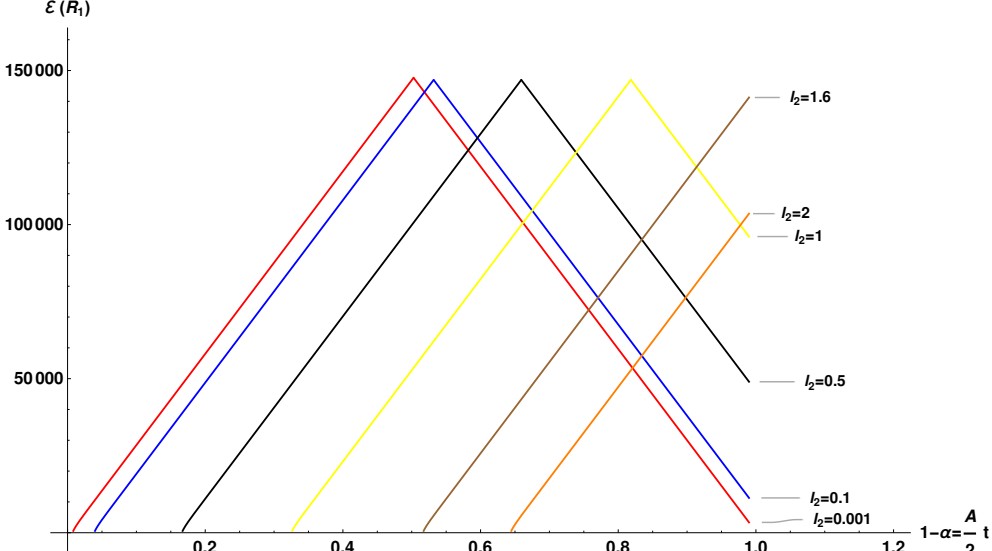

Figure 2: Plot for the behaviour of entanglement negativity of a single interval in a bath as a function of $1-\alpha$ by fixing the length of the rest of the bath denoted as $l_2$. Here, $\Phi_0 = 1000$, $c = 500$, $\beta = .1$, $\phi' = 0.001$ and $b = .001$. Note that when $l_2$ is large we only obtain a part of the Page curve whereas when it is very small we obtain the entire Page curve.

may interpret the plot from the appearance of the entanglement island for the subsystem considered as follows. When the interval considered is smaller than half the size of the entire system, the negativity we obtain rises linearly during the black hole evaporation as the bath subsystem collects more and more Hawking quanta. The entanglement keeps rising because the subsystem considered does not admit the corresponding island for such a configuration. Hence we obtain only the rising part of the analogue of the Page curve for the entanglement negativity of a pure state. However, when the size of the subsystem is larger than half the size of the entire system, the subsystem admits an entanglement island which leads to the purification of the Hawking quanta collected. This in turn leads to a decrease in the entanglement between $R$ and the rest of the system which involves the black hole and the remaining bath. Finally, when the rest of the bath is vanishingly small, the subsystem $R$ spans the entire bath and the corresponding island purifies all the Hawking quanta collected leading to an analogue of the complete Page curve as depicted in figure 2.

### 3.1.2 Adjacent Intervals in Bath

In this subsection we consider the mixed state configuration given by two generic adjacent subsystems $R_1 = [b, b + \ell_1]$ and $R_2 = [b + \ell_1, b + \ell_1 + \ell_2]$ in the bath . The corresponding entanglement negativity between $R_1$ and $R_2$ is given by the conjecture in eq.(20),

$$\mathcal{E} = \frac{1}{2} \left[ S^{(1/2)}(R_1) + S^{(1/2)}(R_2) - S^{(1/2)}(R_1 \cup R_2) \right], \tag{32}$$

where, the Renyi entropies of order half $S^{(1/2)}(R_i)$ are defined in eq. (30). We have denoted the lengths of the subsystems $R_i$ , $i = (1, 2)$ by $\ell_i$. We will examine the behaviour of the entanglement negativity of the adjacent subsystems in the bath coupled to non-extremal JT gravity, as a function of the parameters $\alpha$ and the lengths of the subsystems $\ell_1$ and $\ell_2$. In the following, we will consider different cases by systematically varying the ratio $p = \frac{\ell_1}{\ell_2}$ and $\alpha$ and comment on the qualitative features of the entanglement negativity profiles.

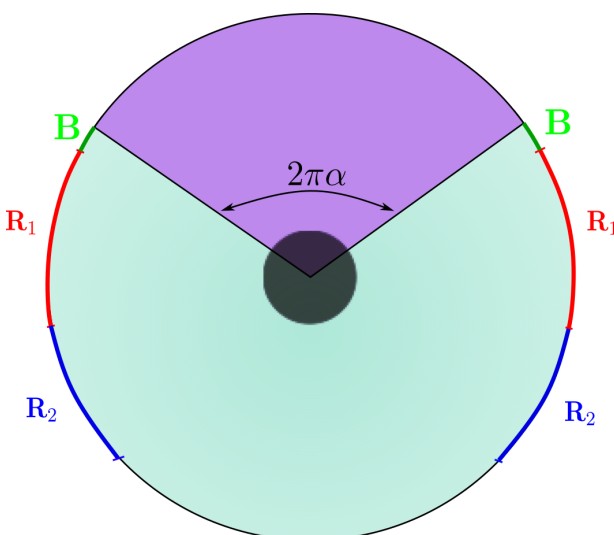

Figure 3: Schematics of two adjacent intervals $R_1$ and $R_2$ in the radiation.

**(a) Keeping $p = \ell_1/\ell_2$ fixed and varying $\alpha$: geometric evaporation**

In this subsection, we obtain the entanglement negativity between the subsystems $R_1$ and $R_2$ which describe two adjacent intervals spanning the entire bath denoted as $R$. This particular situation can be achieved in the limit $b + \ell_1 + \ell_2 \to \pi(1-\alpha)$. As described earlier the dynamical evaporation of the JT black hole is controlled by the value of the parameter $\alpha$. Such geometrical evaporation may be achieved by considering a linearly decreasing dilaton, and the Page time may be identified with $t_{\text{Page}} = -2/\dot{\alpha}(t)$ as explained in [20]. Here, we fix the ratio $p = \frac{\ell_1}{\ell_2}$ and determine the time evolution of the entanglement negativity for the adjacent interval configuration by varying $\alpha$. We may now employ eqs. (30) and (32) to obtain the entanglement negativity of adjacent intervals in bath for the case $p = 1$ as follows

$$\mathcal{E}(R_1,R_2) = \begin{cases} \frac{3}{2}\Phi_0 + \frac{c}{4}\log\left(\frac{\pi^2 \sinh\left[\frac{\pi(2\pi(1-\alpha)-2(b+\ell_1))}{\beta}\right]\sinh^2\left[\frac{\pi\ell_1}{\beta}\right]}{\beta^2 \sinh\left[\frac{\pi(2\pi(1-\alpha)-2b)}{\beta}\right]}\right), & 0 < 1-\alpha < 0.5 \\[4mm] \frac{3}{2}\Phi_0 + \frac{c}{4}\log\left(\frac{\pi^2 \sinh\left[\frac{\pi(2\pi(1-\alpha)-2(b+\ell_1))}{\beta}\right]\sinh^2\left[\frac{\pi\ell_1}{\beta}\right]}{\beta^2 \sinh\left[\frac{\pi(2\pi\alpha+2b)}{\beta}\right]}\right), & 0.5 < 1-\alpha < 1. \end{cases} \tag{33}$$

Similarly, the entanglement negativity for $p = 0.5$ case may be obtained as

$$\mathcal{E}(R_1,R_2) = \begin{cases} \frac{3}{2}\Phi_0 + \frac{c}{4}\log\left(\frac{\pi^2 \sinh^2\left[\frac{\pi\ell_1}{\beta^2}\right]\sinh\left[\frac{\pi(2\pi(1-\alpha)-2(b+\ell_1))}{\beta}\right]}{\beta^2 \sinh\left[\frac{\pi(-2b+2\pi(1-\alpha))}{\beta}\right]}\right), & 0 < 1-\alpha < 0.5 \\[4mm] \frac{3}{2}\Phi_0 + \frac{c}{4}\log\left(\frac{\pi^2 \sinh^2\left[\frac{\pi\ell_1}{\beta}\right]\sinh\left[\frac{\pi(2\pi(1-\alpha)-2(b+\ell_1))}{\beta}\right]}{\beta^2 \sinh\left[\frac{\pi(2b+2\pi\alpha)}{\beta}\right]}\right), & 0.5 < 1-\alpha < \frac{1+p}{2} + \frac{b}{\pi} \\[4mm] \frac{3}{2}\Phi_0 + \frac{c}{4}\log\left(\frac{\pi^2 \sinh^2\left[\frac{\pi\ell_1}{\beta}\right]\sinh\left[\frac{\pi(2\pi\alpha+2(b+\ell_1))}{\beta}\right]}{\beta^2 \sinh\left[\frac{\pi(2b+2\pi\alpha)}{\beta}\right]}\right), & \frac{1+p}{2} + \frac{b}{\pi} < 1-\alpha < 1. \end{cases} \tag{34}$$

In figure 4, we show the plot of the entanglement negativity obtained above with respect to $1-\alpha(t)$ for two different values of $p$. The growth rate along linear rise is $\frac{4c\pi^2}{3\beta}$ for small $\beta << \Phi_r^0$. Remarkably, the plot in figure 4 exactly reproduces an analogue of the Page curve for entanglement negativity for a similar mixed state configuration obtained through the random matrix techniques in [70], which was reviewed in section 2. Hence, our result emphasizes

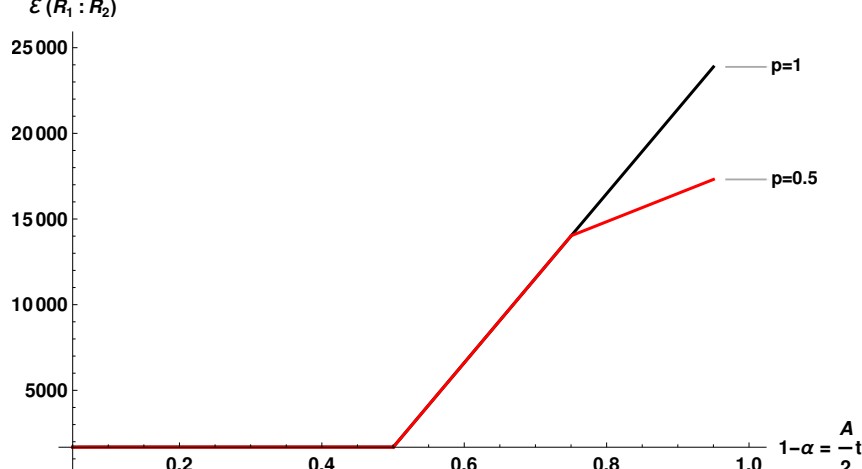

Figure 4: Entanglement negativity between two adjacent subsystems $R_1$ and $R_2$ for $p = 1$ and $p = 0.5$ wrt $(1-\alpha) = \frac{A}{2}t$. Here, $\Phi_0 = 1000$, $c = 500$, $\beta = .1$ and $b = .001$.

the interesting connection between random matrix theory and black hole evaporation, which has been explored recently in relation to diverse phenomena in [5, 54–58, 60, 61]. We discuss different regimes of this plot and comment on the qualitative behaviour of the negativity with respect to the presence of islands for different subregions.

For small values of $(1 - \alpha)$, the black hole $B$ is much larger in size than the bath and therefore the subregions $R_1$ and $R_2$ are strongly entangled with $B$. Therefore, in this regime, the monogamy of entanglement implies a weak entanglement between $R_1$ and $R_2$. Hence, the negativity between them is very small. When the size of $R_1$ or $R_2$ becomes comparable to that of $B$, we land in the tripartite entanglement phase, and the entanglement negativity between $R_1$ and $R_2$ starts to grow linearly in time. There is still one more regime for the entanglement negativity when $p \neq 1$. In such cases, for very small values of $\alpha$, $R_1$ becomes much larger than both $R_2$ and $B$, which corresponds to a maximally entangled phase between $R_1$ and $R_2$. In this regime $R_2$ is completely entangled with $R_1$ and therefore the entanglement negativity between them is determined by the size of the Hilbert space of $R_2$. It will be interesting to investigate more about the transition to this regime.

The above phenomena may be interpreted in terms of the appearance of the negativity islands in the lower dimensional picture. As demonstrated in [48], the quantum extremal surface $Q$ for the entanglement negativity of two generic subsystems $R_1$ and $R_2$ lands on the corresponding entanglement wedge cross section and is given by the intersection of the individual negativity islands for $R_1$ and $R_2$ [7], namely

$$Q = \partial \, \mathrm{Is}_{\mathcal{E}}(R_1) \cap \partial \, \mathrm{Is}_{\mathcal{E}}(R_2). \tag{35}$$

The corresponding entanglement negativity including the island contribution is given as follows

$$\mathcal{E}^{gen}(R_1 : R_2) = \frac{\mathcal{A}^{(1/2)}(Q = \partial \, \mathrm{Is}_{\mathcal{E}}(R_1) \cap \partial \, \mathrm{Is}_{\mathcal{E}}(R_2).)}{4 G_N} + \mathcal{E}^{\mathrm{eff}}(R_1 \cup Is_{\mathcal{E}}(R_1) : R_2 \cup Is_{\mathcal{E}}(R_2)) \,,$$

$$\mathcal{E}(R_1 : R_2) = \min(\mathrm{ext}_Q\{\mathcal{E}^{gen}(R_1 : R_2)\}), \tag{36}$$

---

[7]Incidentally, a similar relationship for the island of the reflected entropy was proposed in [32, 33]. Also note that, we have not made use of eq. (36) to compute the entanglement negativity in our present model.

where $A^{(1/2)}$ corresponds to the area of a backreacted cosmic brane[8] on the EWCS (entanglement wedge cross section) and $\mathcal{E}^{\text{eff}}$ refers to the effective entanglement negativity of quantum matter fields in the bulk regions across the EWCS.

We may now utilize the above mentioned island construction to interpret the plot we obtained in figure 4 for the entanglement negativity. To begin with, the black hole $B$ was very large and the bath subsystems were too small to admit any islands. Correspondingly there is no quantum extremal island described in eq. (35), and the contribution from the effective term in eq. (36) is also very small. As the JT black hole evaporates, the size of the bath subsystems $R_1$ and $R_2$ keeps growing with a fixed ratio between them given by $p$. In such a regime, an island for the entire bath subsystem $R$ appears when it becomes larger than half the size of the total system. This in turn, leads to the appearance of quantum extremal surface for the entanglement negativity of $R_1 \cup R_2$, which shifts towards the boundary as time progresses. As a result the area contribution described by first term in eq. (36) increases and correspondingly the entanglement negativity between $R_1$ and $R_2$ starts increasing linearly in time as they collect more and more Hawking quanta. Note that, depending on the value of the fixed ratio $p$, the nature of the entanglement negativity shows different behaviours. For $p \neq 1$, the larger of the two subsystems $R_1$ and $R_2$ starts developing the corresponding island for entanglement entropy when it becomes larger than half the size of the entire system. When this happens, the rate of growth of the entanglement negativity diminishes as the appearance of the corresponding quantum extremal surface leads to the purification of the Hawking quanta already captured by the subsystem[9]. It is important to note that as the black hole evaporates, the size of $R_1$ and $R_2$ keeps increasing. As a result, in this phase the entanglement negativity increases at a slower rate. However for $p = 1$, the entanglement negativity grows linearly and does not involve any further phase transitions. This may be understood from the fact that for such a case, there are no islands associated with either $R_1$ or $R_2$.

Having obtained the behaviour of the entanglement negativity for the adjacent interval configuration under time evolution, we now determine the same for various configurations by changing the relative size of the subsystems while keeping fixed the parameter $\alpha$, which describes the time dependence.

### (b) Keeping $\alpha$ fixed and varying the ratio $\frac{\ell_1}{\ell_1 + \ell_2}$

Here we compute the entanglement negativity between $R_1$ and $R_2$ where the size of the subsystem $R_1$ is increasing while keeping the size of the JT black hole fixed to less than that of the entire bath subsystem $R = R_1 \cup R_2$. The entanglement negativity can be computed using eqs.

---

[8]To make sense out of the backreacted area $A^{(1/2)}$ the context of JT gravity, we first recall that the area term in the entanglement island formula corresponds to the value of the dilaton field at that point [1, 6, 84]. Now, consider the bulk replica geometry corresponding to the $n$-th Renyi entropy. The backreacted area of a cosmic brane homologous to a subsystem (a point in (1+1)-dimension) is obtained in terms of the value of the dilaton field $\phi^{(n)}$ at the conical singularities, which intrinsically depends on the replica parameter $n$. See, for example, [48, 85] for explicit expressions for the backreacting dilaton. Therefore, the order half area term in eq. (36) corresponds to the analytic continuation of the dilaton $\phi^{(n)}$ to $n \to \frac{1}{2}$.

[9]Note that the entanglement negativity islands of $R_1$ and $R_2$ are related to the entanglement entropy island of $R$ as $Is(R_1 \cup R_2) = \text{Is}_{\mathcal{E}}(R_1) \cup \text{Is}_{\mathcal{E}}(R_2)$ which was described in [48]. A similar relation holds for the islands corresponding to the reflected entropy as discussed in [32].

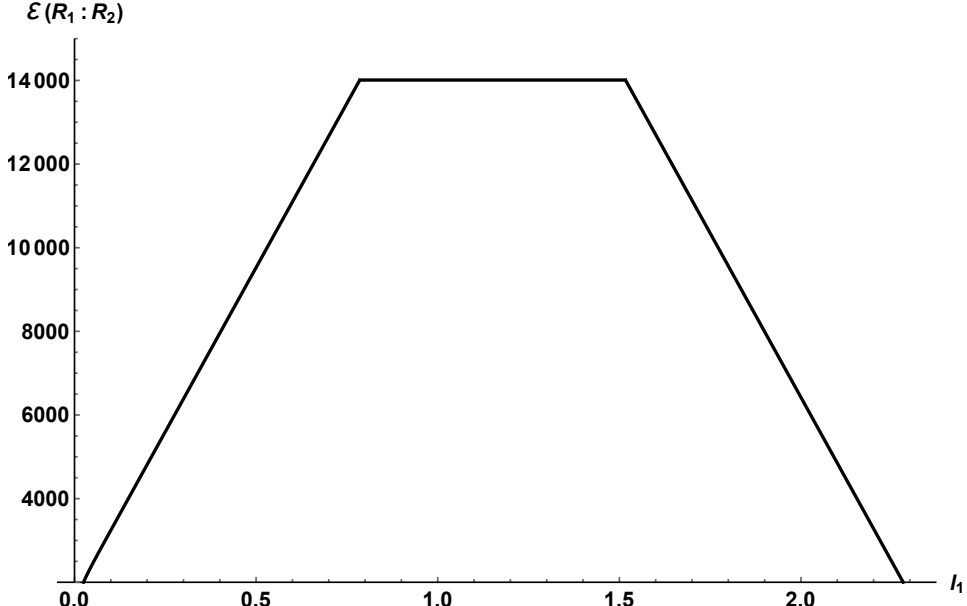

Figure 5: Page curve for entanglement negativity. Here, $\Phi_0 = 1000$, $c = 500$, $\beta = .1$, $\alpha = .25$, $b = .001$ and $b + \ell_1 + \ell_2 = \pi(.99 - \alpha) = 2.325$.

(30) and (32) for the configuration of adjacent intervals as follows

$$\mathcal{E}(R_1, R_2) = \begin{cases} \frac{3}{2}\Phi_0 + \frac{c}{4}\log\left(\frac{\pi^2 \sinh^2\left[\frac{\pi \ell_1}{\beta}\right]\sinh\left[\frac{\pi(2\pi\alpha + 2(b + \ell_1))}{\beta}\right]}{\beta^2 \sinh\left[\frac{\pi(2b + 2\pi\alpha)}{\beta}\right]}\right), & 0 < \ell_1 < \frac{\pi}{2}(1 - 2\alpha) - b \\ \frac{3}{2}\Phi_0 + \frac{c}{4}\log\left(\frac{\pi^2 \sinh^2\left[\frac{\pi \ell_1}{\beta}\right]\sinh\left[\frac{\pi(2\pi(1-\alpha) - 2(b + \ell_1))}{\beta}\right]}{\beta^2 \sinh\left[\frac{\pi(2b + 2\pi\alpha)}{\beta}\right]}\right), & \frac{\pi}{2}(1 - 2\alpha) - b < \ell_1 < \frac{\pi}{2} \\ \frac{3}{2}\Phi_0 + \frac{c}{2}\log\left(\sinh\left[\frac{\pi(2\pi(1-\alpha) - 2(b + \ell_1))}{\beta}\right]\right), & \frac{\pi}{2} < \ell_1 < \pi(1 - \alpha) - b. \end{cases} \quad (37)$$

We plot the entanglement negativity from the above expression in figure 5 which depicts the analogue of the Page curve for the entanglement negativity. The growth rate along linear rise is $\frac{4c\pi}{3\beta}$ for very small $\beta$. Remarkably, the behaviour of the entanglement negativity we obtain from our computations once again matches exactly with that derived in the context of random matrix theory in [70].

In the present setup, the bath captures all of the radiation coming out of the black hole and therefore we are always in the NPT (negative partial transpose) phase and correspondingly the entanglement negativity between $R_1$ and $R_2$ is non-zero. As long as $R_1$ remains smaller than $R_2$ the negativity rises linearly since the entanglement is governed by the size of the smaller subsystem. When we keep on increasing $\ell_1$, at a particular point the size of $R_1$ becomes comparable to that of $R_2$ and the entanglement between them saturates to a plateau, describing a tripartite entanglement phase. Finally, when $R_1$ becomes much larger than $R_2$, the entanglement between them is determined by the size of $R_2$. This leads to a linearly decreasing phase of the entanglement negativity when plotted as a function of $\ell_1$. This may observed from the plot in figure 5 .

In the two-dimensional point of view these transitions may be understood in terms of the appearance of the entanglement islands. As described before, in this scenario, the JT black hole is smaller than half the size of the total system and therefore the radiation bath $R = R_1 \cup R_2$ always accommodates its entanglement island. When the size of the bath subsystem $R_1$ is small, then $R_2$ admits an island as one of the geodesics corresponding to $R_2$ passes through

the JT gravity region. As we increase the size of $R_1$, due to a decrease in the size of $R_2$ this geodesic moves downwards and therefore the size of the entanglement island for $R_2$ decreases. This in turn causes a decrease in the purification of the Hawking quanta present in $R_2$. As a result the negativity between $R_1$ and $R_2$ increases due to the increasing size of $R_1$ as well as a decreasing size of the island for $R_2$. If we further increase the size of $R_1$, at a particular point $R_2$ becomes smaller than half the size of the entire system $R \cup B$ (black hole+bath). Hence, its island disappears completely and correspondingly we land in the tripartite entanglement phase. In this regime, the entanglement negativity saturates to a plateau since the total number of entangling modes between $R_1$ and $R_2$ remains fixed. Finally, $R_1$ admits its own entanglement island when it becomes larger than half the size of the entire system. As a result, the entangling modes captured in $R_1$ gets purified from their partners collected in the corresponding island. This leads to a net decrease of the available entangling modes in $R_1$ with both $R_2$ and $B$. Note that as the size of $R_2$ decreases, the entanglement negativity between $R_1$ and $R_2$ also decreases linearly until it vanishes completely.

**(c) Keeping $\alpha$ and $\ell_1$ fixed, varying $\ell_2$**

To begin with, we keep $\alpha$ and $\ell_1$ fixed, and increase $\ell_2$ till it covers the full radiation region. Note that as described earlier the parameter $\alpha$ controls the time dependence in the present construction. Hence, fixing it describes a particular time frozen situation during the black hole evaporation, when a fixed amount of Hawking radiation has been transferred from the black hole to the bath. We then vary the size $\ell_2$ of the subsystem $R_2$ and investigate the behaviour of entanglement negativity for the bipartite system $R_1 \cup R_2$. The entanglement negativity may now be computed by utilizing the expression for Renyi entropy of order half given in (30) for subsystems $R_1, R_2$ and $R_1 \cup R_2$ in eqs. (32) as follows

$$
\mathcal{E}(R_1, R_2) = \begin{cases} \frac{3}{2}\Phi_0 + \frac{c}{2}\log\left( \frac{\pi \sinh\left[\frac{\pi\ell_1}{\beta}\right]\sinh\left[\frac{\pi\ell_2}{\beta}\right]}{\beta \sinh\left[\frac{\pi(\ell_1+\ell_2)}{\beta}\right]} \right), & \ell_2 < \frac{\pi}{2} - \ell_1 \\[3ex] \frac{3}{2}\Phi_0 + \frac{c}{4}\log\left( \frac{\pi^4 \sinh^2\left[\frac{\pi\ell_1}{\beta}\right]\sinh^2\left[\frac{\pi\ell_2}{\beta}\right]}{\beta^4 \sinh\left[\frac{\pi(2b+2\pi\alpha)}{\beta}\right]\sinh\left[\frac{\pi(2\pi(1-\alpha)-2(b+\ell_1+\ell_2))}{\beta}\right]} \right), & \frac{\pi}{2} < \ell_2 < \frac{\pi}{2} - \ell_1 \\[3ex] \frac{3}{2}\Phi_0 + \frac{c}{4}\log\left( \frac{\pi^2 \sinh^2\left[\frac{\pi\ell_1}{\beta}\right]\sinh\left[\frac{\pi(2\pi\alpha+2(b+\ell_1))}{\beta}\right]}{\beta^2 \sinh\left[\frac{\pi(2b+2\pi\alpha)}{\beta}\right]} \right), & \ell_2 > \frac{\pi}{2}. \end{cases}
\tag{38}
$$

In the following we discuss and qualitatively substantiate the behaviour of the entanglement negativity in different regimes as sketched in fig. 6. The growth rate along linear rise is $\frac{4c\pi}{3\beta}$ for high temperatures. The authors of [20] argued that the region bounded by the RT surface(s) for a generic subsystem in the bath, and the dividing line of the JT and BTZ regions may be identified as the corresponding island. In the present model, therefore, the transition from the island to the no island phase is mimicked by the transition of the geodesic from within the purple region to the exterior. We will utilize these facts to make qualitative comments about how the nature of the entanglement negativity changes with the (dis)appearance of islands.

Note that as we vary $\ell_2$, the entanglement negativity is almost vanishingly small upto a certain size of $R_2$ for a fixed size of $R_1$ and $\alpha$. As entanglement negativity is characterized by the negative eigenvalues of $\rho_R^{T_2}$, its vanishing may be interpreted as $\rho_R$ being a "PPT" (positive partial transpose) state [70][10]. As long as the dimension of the Hilbert space of $R = R_1 \cup R_2$ remains much smaller than that of its complement, the density matrix corresponding to the

---

[10]Note that the PPT criterion is a necessary condition for a state to be separable but it is sufficient only for $2 \times 2$ and $2 \times 3$ dimensional Hilbert spaces. However, for higher dimensional Hilbert spaces it remains only as a necessary condition. Hence, the vanishing entanglement negativity implies that there is no distillable entanglement in a given state but it leaves out a class of entangled states known as the *bound entangled states* [34, 39, 86, 87].

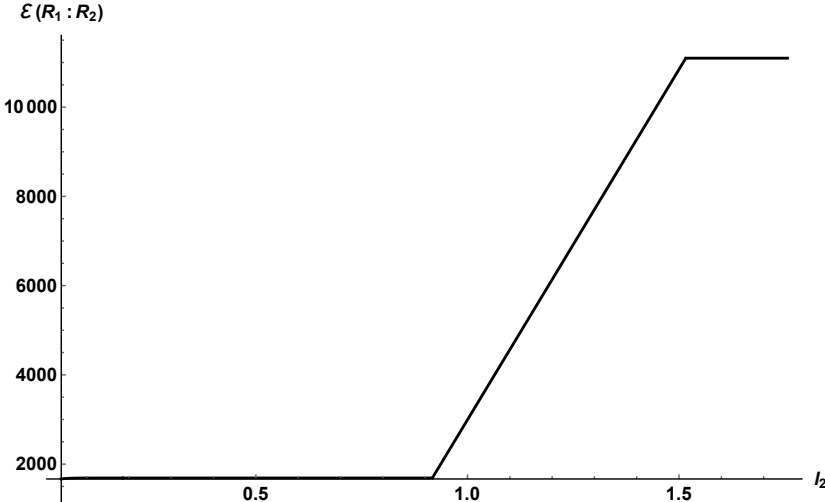

Figure 6: Entanglement negativity between $R_1$ and $R_2$ w.r.t the size of $R_2$. Here, $\Phi_0 = 1000$, $c = 500$, $\beta = .1$, $b = .001$, $b + \ell_1 = .6$ and $\alpha = 0.25$.

radiation remains in a PPT state. This may be interpreted as the subsystems $R_1$ and $R_2$ being too small compared to the rest, to have any entanglement between each other [11].

As we further increase $\ell_2$, the Hilbert space of $R$ becomes larger than its complement and more and more Hawking quanta are collected in the bath subsystem $R_2$. These quanta are in turn entangled with the quanta collected in the bath subsystem $R_1$. Hence, in this regime, $\rho_R^{T_2}$ gains negative eigenvalues and correspondingly the entanglement negativity between $R_1$ and $R_2$ starts to increase linearly with $\ell_2$.

Finally, when $\ell_2$ becomes very large, $R$ captures most of the Hawking radiation coming out of the black hole $B$. As a result $R_1$ and $R_2$ both become entangled with $B$. This is the tripartite entanglement phase, where the entanglement negativity between $R_1$ and $R_2$ saturates to a plateau. This may be understood as follows: when $R_2$ becomes larger than half of the entire system then the entanglement between $R_1$ and $R_2$ reaches its maximum. This is because all the Hawking quanta in $R_1$ are entangled with their partners in $R_2$ and their entanglement with each other cannot grow any further.

One may utilize the above island formulation for entanglement negativity to differentiate its behavior in different regimes depicted in fig. 6. In the first case there is no entanglement islands for $R_1$, $R_2$ and $R$, and therefore the area term in eq. (35) vanishes. As described above, due to smaller sizes of $R_1$ and $R_2$, the bulk (effective) term in eq. (36) also remains vanishingly small.

Note that depending on the (fixed) size of $R_1$, it might or might not admit an entanglement island. Let us first consider the case when $R_1$ does not have an island. In this case initially the entanglement negativity vanishes due to the arguments described above. As we increase the size of the subsystem $R_2$, at a particular point the subsystem $R$ will be greater than half of the entire system. At this point it will be large enough to accommodate an entanglement island and the RT surface corresponding to it transits from the disconnected phase to the connected one which is depicted in fig. 7. This leads to the appearance of the quantum extremal surface for the entanglement negativity, which is the shared boundary of the individual negativity islands $\text{Is}_{\mathcal{E}}(R_1)$ and $\text{Is}_{\mathcal{E}}(R_2)$ as described in eq. (35)[12]. In this regime, with increasing

---

[11]Note that in fig. 6, in this regime the entanglement negativity has some small finite value due to the finiteness of the UV cut-off $\epsilon$.

[12]Note that in this phase $R_1$ and $R_2$ do not admit their entanglement entropy islands as either of them is smaller than half of the entire system. However they admit entanglement negativity islands as the combined system $R_1 \cup R_2$

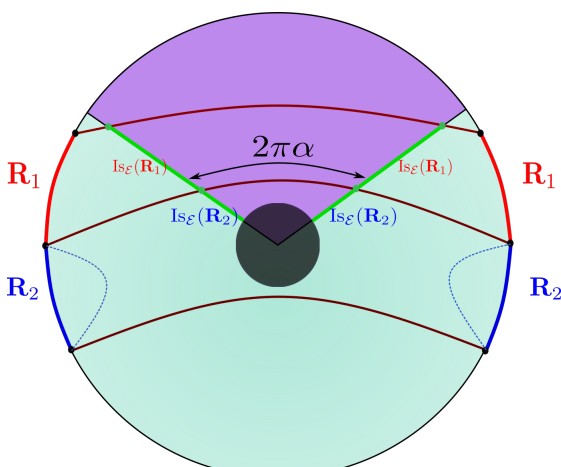

Figure 7: Entanglement entropy and negativity Islands corresponding to the two adjacent intervals in the radiation. The full green segments correspond to the entanglement entropy islands of the subsystem $R = R_1 \cup R_2$. Note that the dashed curves correspond to the disconnected geodesics for the subsystem $R_2$.

$\ell_2$, the corresponding negativity island $Is_{\mathcal{E}}(R_2)$ increases in size. As a result the entanglement negativity island given in eq. (35) shifts towards the asymptotic boundary along the radial direction. This leads to a linear increase [13] in the dominant area term in eq. (36) and therefore, the entanglement negativity between $R_1$ and $R_2$ increases as shown in fig. 6.[14] As we further increase $\ell_2$, the subsystem $R_2$ starts claiming its own entanglement island as shown in fig. 8, and the RT surface corresponding to it transitions from the disconnected phase to the connected phase. In this regime, the Hawking quanta already accommodated by $R_2$ find their partners in the corresponding entanglement entropy island of $R_2$ and therefore a purification happens. As a result we obtain the plateau region as shown in fig. 6 indicating entanglement saturation. Next let us consider the case when $R_1$ admits an entanglement island. In this case, the behavior is similar to the earlier case except that the vanishing phase of the entanglement negativity disappears. This is because in this case, the subsystems $R_1$ and hence $R$ always admit their entanglement islands which lead to the non-vanishing entanglement negativity which may be expressed as

$$
\mathcal{E}(R_1, R_2) = \begin{cases} \frac{3}{2}\Phi_0 + \frac{c}{4}\log\left(\frac{\pi^2 \sinh^2\left[\frac{\pi\ell_2}{\beta}\right] \sinh\left[\frac{\pi(2\pi(1-\alpha)-2(b+\ell_1))}{\beta}\right]}{\beta^2 \sinh\left[\frac{\pi(2\pi(1-\alpha)-2(b+\ell_1+\ell_2))}{\beta}\right]}\right), & \ell_2 < \frac{\pi}{2} \\ \frac{3}{2}\Phi_0 + \frac{c}{2}\log\left(\sinh\left[\frac{\pi(2\pi(1-\alpha)-2(b+\ell_1))}{\beta}\right]\right), & \text{otherwise.} \end{cases}
\tag{39}
$$

The behavior of the entanglement negativity for this case is depicted in fig. 9. The growth rate along linear rise is $\frac{4c\pi}{3\beta}$ for very small $\beta$.

---

admits entanglement entropy island.

[13]The linear increase in the area term may be understood from the dilaton profile in eq. (10). Utilizing the form of the tortoise coordinate $r^*$ given in [20] the dilaton profile in the Schwarzschild coordinates may be seen to be given by $\Phi(r) = \Phi_0 + \Phi_r\, r$. Therefore, as one moves towards the asymptotic boundary of the spacetime, the dilaton and hence the area increases linearly.

[14]One can understand the linear rise of the entanglement negativity in this region in terms of redistribution of entanglement between Hawking quantas in the corresponding entanglement negativity islands. In the lower dimensional effective theory one should consider $R_1 \cup Is_{\mathcal{E}}(R_1)$ and $R_2 \cup Is_{\mathcal{E}}(R_2)$ as the entangling subsystems and therefore the effective entanglement negativity decreases due to a partial purification of available entanglement between Hawking modes present in the subsystems. However the total entanglement negativity shows a linear rise because of the appearance of a shared boundary in between $Is_{\mathcal{E}}(R_1)$ and $Is_{\mathcal{E}}(R_2)$ across which a large number of modes are entangled.

![Sci|Post]

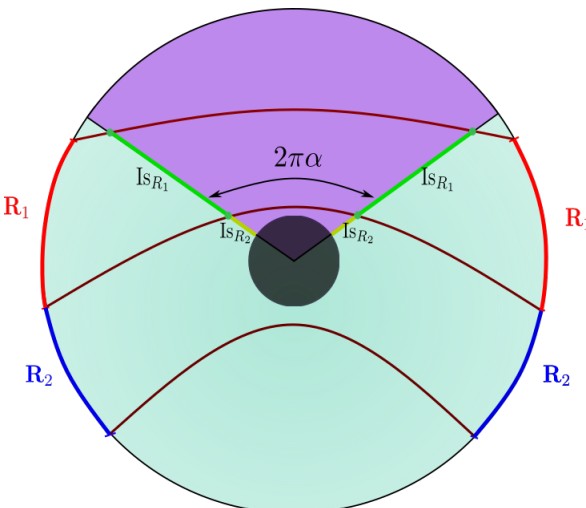

Figure 8: The subsystem $R_2$ is big enough to claim its own entanglement entropy island. This corresponds to connected RT surfaces for $R_2$.

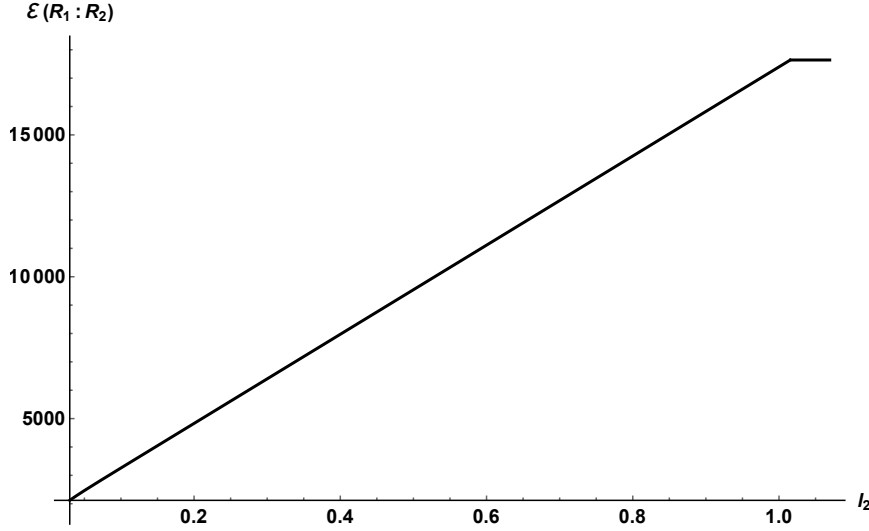

Figure 9: Schematic behaviour of entanglement negativity as a function of $\ell_2$ in the presence of an entanglement entropy island for $R_1$. Here, $\Phi_0 = 1000$, $c = 500$, $\beta = .1$, $\alpha = .15$, $b = .001$ and $b + \ell_1 = 1.6$.

### 3.1.3 Disjoint Intervals in Bath

Having discussed the case of adjacent intervals we now analyze the entanglement negativity of a mixed state configuration of two disjoint subsystems described by the intervals $R_1 = [b, b + \ell_1]$ and $R_2 = [b + \ell_1 + \ell_s, b + \ell_1 + \ell_s + \ell_2]$ and $R_s = [b + \ell_1, b + \ell_1 + \ell_s]$ denotes the interval sandwiched in between $R_1$ and $R_2$ as shown in fig. 10. The expression for the holographic entanglement negativity for two disjoint subsystems in proximity in the $CFT_2$ bath are given in eq.(22). Here the length of the subsystem $R_i$ is denoted by $\ell_i$. In this subsection we consider the same model as in section 3.1.2 and explore different situations involving various limit of the subsystems $R_1,R_2$ and $R_s$. First we keep the size of the subsystems $R_1$ and $R_s$ fixed and increase the size of the subsystem $R_2$. In the second case we keep $R_1$ fixed and vary the relative size between the subsystems $R_2$ and $R_s$ by changing their shared boundary point. The entanglement negativity in the first scenario shows behaviour similar to that of the

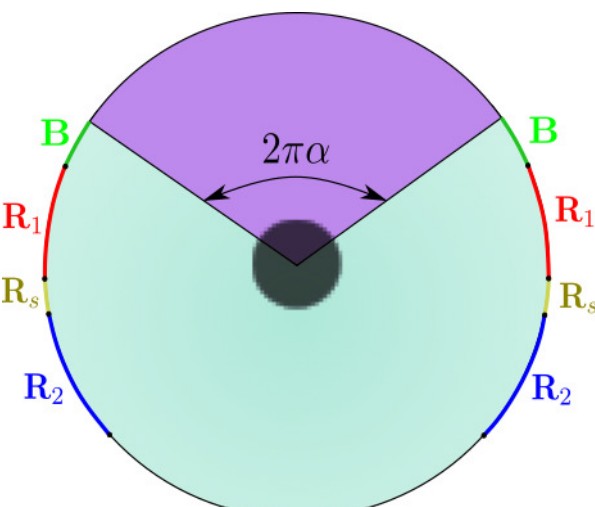

Figure 10: Schematics of two disjoint intervals $R_1$ and $R_2$ in the bath sandwiching the subsystem $R_s$.

previous case for two adjacent intervals in the bath discussed in section 3.1.2(a). The result in the second case also agrees with the physical interpretation described in the section 3.1.2.

**(a) Keeping $\ell_1$, $\ell_s$ fixed and changing $\ell_2$**

To begin with, we keep $\ell_1$, $\ell_s$ and the size of the JT black hole fixed and increase $\ell_2$ till the subsystem $R_2$ covers the whole radiation. The entanglement negativity can be computed for this configuration using eqs. (22) and (30) as follows

$$
\mathcal{E}(R_1, R_2) =
\begin{cases}
\frac{c}{2} \log\left( \frac{\sinh\left[\frac{\pi(\ell_1+\ell_s)}{\beta}\right] \sinh\left[\frac{\pi(\ell_2+\ell_s)}{\beta}\right]}{\sinh\left[\frac{\pi\ell_s}{\beta}\right] \sinh\left[\frac{\pi(\ell_1+\ell_2+\ell_s)}{\beta}\right]} \right), & \ell_2 < (\frac{\pi}{2} - \ell_1 - \ell_s) \\[2em]
\frac{c}{4} \log\left( \frac{\pi^2 \sinh^2\left[\frac{\pi(\ell_1+\ell_s)}{\beta}\right] \sinh^2\left[\frac{\pi(\ell_2+\ell_s)}{\beta}\right]}{\beta^2 \sinh^2\left[\frac{\pi\ell_s}{\beta}\right] \sinh\left[\frac{\pi(2b+2\pi\alpha)}{\beta}\right] \sinh\left[\frac{\pi(2\pi(1-\alpha)-2(b+\ell_1+\ell_2+\ell_s))}{\beta}\right]} \right), & \\[0.5em]
& (\frac{\pi}{2} - \ell_1 - \ell_s) < \ell_2 < (\frac{\pi}{2} - l_s) \\[2em]
\frac{c}{4} \log\left( \frac{\sinh^2\left[\frac{\pi(\ell_1+\ell_s)}{\beta}\right] \sinh\left[\frac{\pi(2\pi\alpha+2(b+\ell_1))}{\beta}\right]}{\sinh^2\left[\frac{\pi\ell_s}{\beta}\right] \sinh\left[\frac{\pi(2b+2\pi\alpha)}{\beta}\right]} \right), & \ell_2 > (\frac{\pi}{2} - l_s)
\end{cases}
\tag{40}
$$

The behaviour of the above entanglement negativity with increasing $\ell_2$ is shown in fig. 11 where we have considered $b = 0.001$, $\ell_1 = 0.6$, $\ell_s = 0.4$ and $\alpha = 0.25$. In the following we explain the different regimes of fig. 11 and also interpret these phase transitions using the island arguments in the effective lower dimensional scenario. The growth rate along linear rise is $\frac{c\pi}{\beta}$ for high temperatures ( for small $\beta$).

As shown in fig. 11 the entanglement negativity between $R_1$ and $R_2$ retains an infinitesimally small value till the size of the entire bath subsystem $R \equiv R_1 \cup R_s \cup R_2$ is less than half the size of the entire system. In this region $R_1$ and $R_2$ can not have any entanglement as all the Hawking quanta in the region $R_1 \cup R_s \cup R_2$ are entangled with the rest of the system. If we keep increasing $\ell_2$, then we land in a phase where the size of the entire bath subsystem $R$ is larger than half the size of the full system $R \cup B$. When this happens, the subsystems $R_1$ and $R_2$ gather enough Hawking quanta, some of which lead to their entanglement with the rest of the system whereas the remaining quanta result in their mutual entanglement. Hence, the entanglement negativity shows a linear rise with increasing $\ell_2$. It can be seen in fig. 11 that

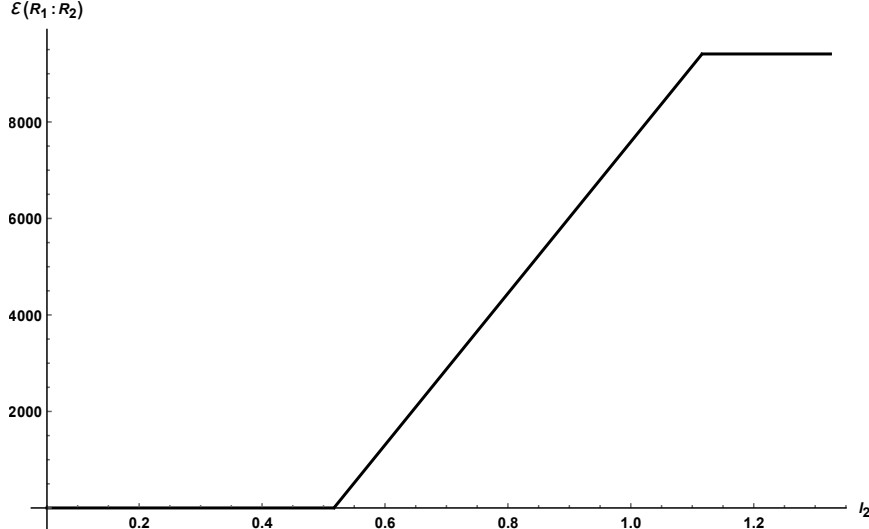

Figure 11: Entanglement negativity between two disjoint intervals ($R_1 = [b, b+\ell_1]$ and $R_2 = [b+\ell_1+\ell_s, b+\ell_1+\ell_s+\ell_2]$) in proximity in the bath as a function of $\ell_2$. In this case, we have chosen $\Phi_0 = 1000$, $c = 500$, $\beta = .1$, $b = 0.001$, $\ell_1 = 0.6$, $\ell_s = 0.4$ and $\alpha = 0.25$.

the linear rise ceases when the size of the subsystem $R_2 \cup R_s$ becomes larger than half the size of the entire system. Beyond this point the entanglement negativity between $R_1$ and $R_2$ shows a plateau region which indicates a constant entanglement between $R_1$ and $R_2$ despite the increasing size of the subsystem $R_2$. In this region, $R_1$ being a part of $(R_2 \cup R_s)^c$, is completely entangled with the subsystem $(R_2 \cup R_s)$. As a result, $R_1$ does not have any more Hawking quanta left which could lead to a rise in its entanglement with $R_2$ even if $\ell_2$ is increased further. The phenomenon described above can be explained in terms of the negativity islands from the two-dimensional point of view along the same line as explained in section 3.1.2. The initial flat region in fig. 11 corresponds to the situations where $R \equiv R_1 \cup R_s \cup R_2$ is smaller than the half of the entire system and thus it does not have any entanglement entropy island. As a result the entanglement negativity picks up the contribution only from the effective term in eq.(36) which is vanishingly small. In the second phase, the size of the subsystem $R$ is larger than the rest of the system and therefore $R$ develops its own entanglement entropy island. Hence, the entanglement negativity islands for $R_1$ and $R_2$ appear with a common boundary whose area contributes to the entanglement negativity. As we increase the size of $R_2$ the size of the entanglement negativity island corresponding to $R_2$ increases and therefore the entanglement negativity between $R_1$ and $R_2$ increases linearly as they collect more and more Hawking quanta (cf. the discussion after fig. 7). Now as we further increase $\ell_2$ the subsystem $R_2$ admits its own entanglement entropy island when the size of $R_2$ becomes larger than half the size of the total system. This new entanglement entropy island collects the partners of the Hawking quanta present in $R_2$ which leads to a purification. As a result the entanglement negativity becomes constant in this region with the increasing size of $R_2$.

**(b) Keeping $\ell_1$ fixed, varying $\ell_2$ and $\ell_s$**

Next we keep $\alpha$, $\ell_1$ and the size of the entire bath system $R$ fixed while we vary the size of the intermediate sub-region $R_s$. We choose to keep the size of $R_1$ small enough such that for smaller $R_s$ (larger $R_2$), $\rho_{R_1 \cup R_2}$ is in the NPT phase and therefore the entanglement negativity between $R_1$ and $R_2$ remains non-zero. Now the entanglement negativity for this case may be

obtained using eqs. (22) and (30) as

$$\mathcal{E}(R_1,R_2) = \begin{cases} \frac{c}{4}\log\left(\frac{\sinh^2\left[\frac{\pi(\ell_1+\ell_s)}{\beta}\right]\sinh\left[\frac{\pi(2\pi\alpha+2(b+\ell_1))}{\beta}\right]}{\sinh^2\left[\frac{\pi\ell_s}{\beta}\right]\sinh\left[\frac{\pi(2b+2\pi\alpha)}{\beta}\right]}\right), & \ell_s < (\frac{\pi}{2}-\ell_1) \\[4mm] \frac{c}{4}\log\left(\frac{\sinh\left[\frac{\pi(2\pi\alpha+2(b+\ell_1))}{\beta}\right]\sinh\left[\frac{\pi(2\pi(1-\alpha)-2(b+\ell_1+\ell_s))}{\beta}\right]}{(\pi/\beta)^2\sinh^2\left[\frac{\pi\ell_s}{\beta}\right]}\right), & (\frac{\pi}{2}-\ell_1) < \ell_s < \frac{\pi}{2} \\[4mm] 0, & \ell_s > \frac{\pi}{2}. \end{cases} \tag{41}$$

We observe from the above expression that the entanglement negativity remains constant upto a certain size of $R_s$ and then linearly decreases to zero which is plotted in fig. 12.

As we increase $\ell_s$, with a fixed endpoint of $R_2$ the size of $R_2$ gradually decreases. As long as $R_2$ remains larger than half the entire system, we are in the tripartite entanglement phase and the entanglement negativity between $R_1$ and $R_2$ remains constant. After $\ell_s$ crosses a certain threshold, $R_2$ becomes smaller than half of the entire system and the entanglement negativity starts decreasing linearly. Finally, when $R_1$ and $R_2$ are small enough to be maximally entangled with the rest of the system $B \cup R_s$, they do not have any entanglement with each other due to the monogamy property. In this regime, the entanglement negativity becomes vanishingly small.

From the point of view of the two-dimensional effective theory, the full bath subsystem $R = R_1 \cup R_s \cup R_2$ always has an entanglement entropy island for this particular configuration. For smaller $R_s$, $R_2$ is large enough to admit its own entanglement island. As a result, some of the Hawking quanta collected by $R_2$ are purified by their partners in the entanglement island and the entanglement negativity between $R_1$ and $R_2$ remains constant. As we increase $\ell_s$, after a certain point the entanglement entropy island of $R_2$ disappears. However, note that since $R$ has an entanglement entropy island, there is a quantum extremal surface corresponding to the entanglement negativity islands of $R_1$ and $R_2$ as described by the relation in the footnote 12. With increasing $\ell_s$, or a decreasing $\ell_2$ the above mentioned quantum extremal surface shifts towards the center and therefore the area contribution in eq. (36) decreases linearly

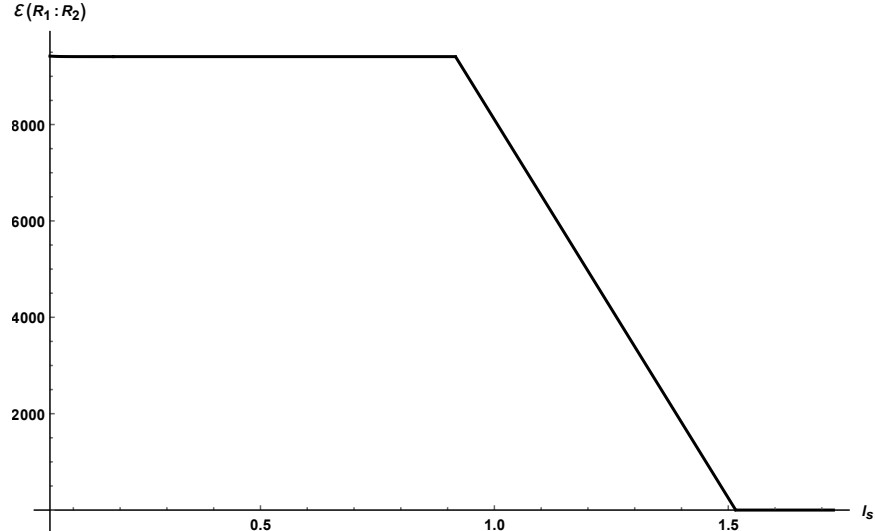

Figure 12: Variation of the entanglement negativity for two disjoint intervals $R_1$ and $R_2$ with increasing size of the intermediate interval $R_s$. Here we choose the values $\Phi_0 = 1000$, $c = 500$, $\beta = .1$, $b = 0.001$, $b + \ell_1 = 0.6$, $b + \ell_1 + \ell_s + \ell_2 = 2.325$ and $\alpha = 0.25$.

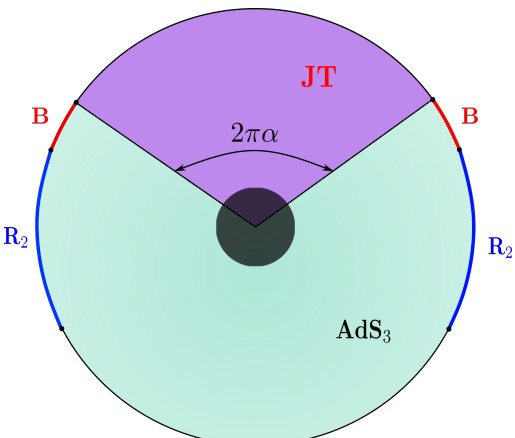

Figure 13: Schematics of two adjacent intervals $B$ and $R_2$ involving the JT black hole and the bath.

leading to a decrease in the entanglement negativity between $R_1$ and $R_2$. Finally, for a very large $R_s$, the entanglement wedge between $R_1$ and $R_2$ is disconnected and correspondingly the negativity island disappears completely. In this phase, the entanglement negativity only picks up the effective contribution in eq. (36), which due to the relatively small sizes of $R_1$ and $R_2$, is vanishingly small.

### 3.1.4 Adjacent Intervals involving Black hole and Bath

In this subsection we consider two adjacent intervals $B = [0, b]$ and $R_2 = [b, b + \ell_1]$, where the interval $B$ includes the quantum mechanical degrees of freedom as shown in fig. 13. We interpret this configuration as two adjacent subsystems involving the black hole and the bath. We vary the size of the bath subsystem $R_2$ and plot the entanglement negativity corresponding to this configuration in fig. 14. The behaviour of the entanglement negativity between $B$ and $R_2$ is similar to the situation described in subsection 3.1.2 (a) and is depicted in fig. 6. The qualitative features of the entanglement negativity in the different regimes may be explained using arguments similar to those in subsection 3.1.2.

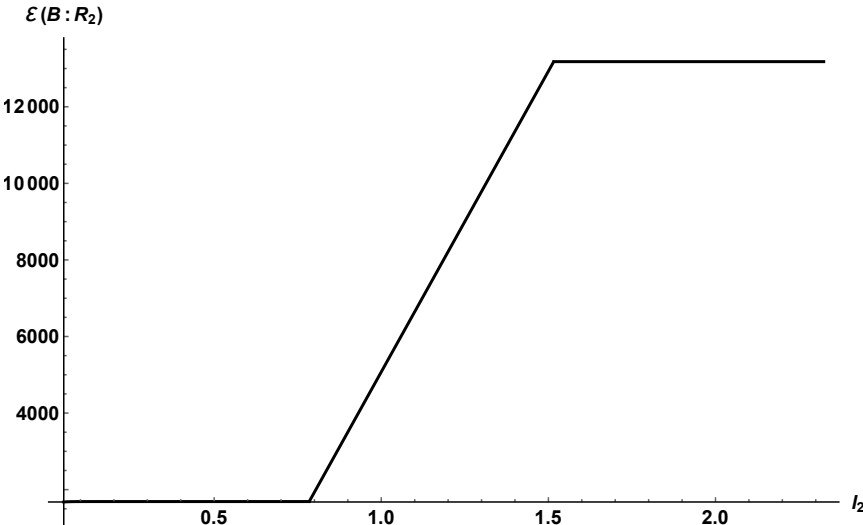

Figure 14: Variation of the entanglement negativity between the adjacent intervals $B$ and $R_2$, with the size of $R_2$. Here $\Phi_0 = 1000$, $c = 500$, $\beta = .1$, $b = 0.001$, $\alpha = 0.25$.

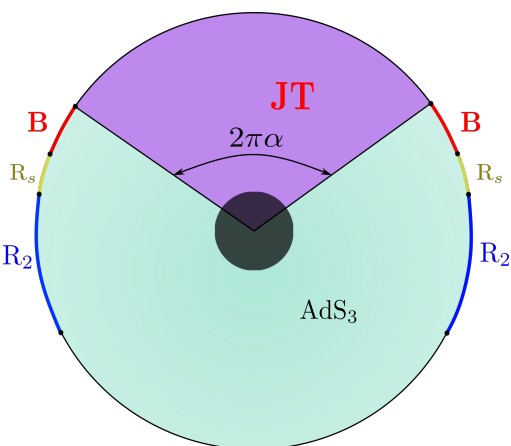

Figure 15: Schematics of two disjoint intervals $B$ and $R_2$ involving the JT black hole and the bath.

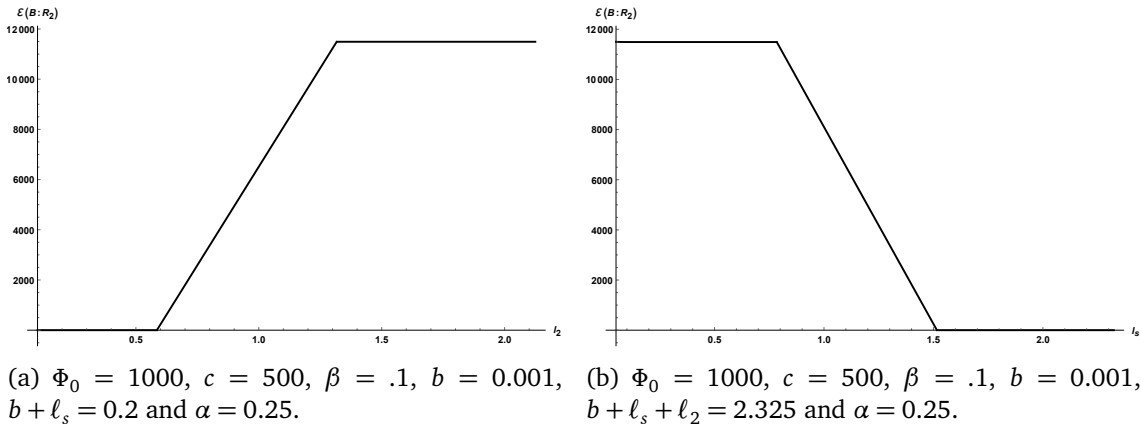

(a) $\Phi_0 = 1000$, $c = 500$, $\beta = .1$, $b = 0.001$, $b + \ell_s = 0.2$ and $\alpha = 0.25$.

(b) $\Phi_0 = 1000$, $c = 500$, $\beta = .1$, $b = 0.001$, $b + \ell_s + \ell_2 = 2.325$ and $\alpha = 0.25$.

Figure 16: Entanglement negativity between two disjoint intervals $B = [0, b]$ and $R_2 = [b + \ell_s, b + \ell_s + \ell_2]$ involving both the JT black hole and the radiation bath. In (a) the size of the radiation subsystem $R_2$ is varied while in (b) we vary the size of the intermediate subsystem $R_s$.

### 3.1.5 Disjoint Intervals involving Black hole and Bath

Finally we consider the mixed state configuration of two disjoint intervals involving the JT black hole and the bath. The configuration is schematically shown in fig. 15 and is similar to that in fig. 10, only in this case the subsystem $B = [0, b]$ includes the quantum mechanical degrees of freedom while $R_s = [b, b + \ell_s]$ and $R_2 = [b + \ell_s, b + \ell_s + \ell_2]$ reside in the radiation bath. As in subsection 3.1.3, we consider two cases. In the first case we keep the size of the JT black hole fixed and vary $\ell_2$ while keeping $\ell_s$ fixed. In the second case we vary $\ell_s$ keeping the size of the black hole and one of the endpoints of $R_2$ fixed. The plots of the entanglement negativity between $B$ and $R_2$ are shown in fig. 16. Once again the qualitative features of these plots may be explained from the lower dimensional island arguments and also in terms of the Hawking quanta collected by the individual subsystems, similar to those in subsection 3.1.3.

# 4 Summary and discussion

To summarize, in this article we have obtained the behaviour of the entanglement negativity for various bipartite pure and mixed state configurations involving adjacent, disjoint and single intervals in a bath coupled to an evaporating non-extremal JT black hole obtained through the partial dimensional reduction of a BTZ black hole as described in [20]. Note that in the present article we utilized the model developed in [20] that geometrizes the black hole evaporation and the island formation in the two dimensional effective theory. We would like to emphasize that the Hawking radiation collected in the two dimensional bath is entangled with the evaporating JT black hole, and hence probing its entanglement structure requires a mixed state entanglement measure such as the entanglement negativity. Hence, our investigation of the behaviour of the entanglement negativity for various configurations essentially amounts to exploring the dynamics of the mixed state entanglement structure of the Hawking radiation collected in a bath coupled to an evaporating JT black hole. Remarkably, we have demonstrated that the results we obtained exactly reproduced the analogues of Page curve for the entanglement negativity in bipartite random mixed states, which was derived through a diagrammatic technique in [70]. Therefore, our computation reiterates the deep connection between black holes and random matrix models which has been investigated in several recent articles in diverse contexts [5, 54–62].

We started by considering the entanglement negativity of a bipartite pure state configuration involving a single interval in the bath and the rest of the system. For this case, the entanglement negativity is given by the Renyi entropy of order half as expected from quantum information theory. We demonstrated that when the single interval considered is restricted to be less than half the size of the entire system, the behaviour of the entanglement negativity is analogous to the rising part of the Page curve for the entanglement entropy. However, when the interval considered becomes larger than half the size of the entire system the entanglement negativity starts decreasing and finally when it spans the entire bath we obtain the complete Page curve. We interpreted the above result in terms of the entanglement island construction in the two dimensional effective theory as follows. An entanglement island for this case appears only when the size of the bath subsystem under consideration is larger than half the size of the entire system. This leads to the purification of the Hawking quanta collected in the subsystem resulting in a net decrease of the entanglement negativity. When the subsystem considered spans the entire bath all the Hawking quanta are purified by the corresponding island and we obtain the analogue of the complete Page curve for the pure state.

We then considered the mixed state configurations involving the adjacent and the disjoint intervals in the bath. We further divided the case of the adjacent intervals in the bath into three. In the first scenario, we kept the ratio of the two adjacent intervals fixed and varied the dimensional reduction parameter $\alpha$ which controls the geometric evaporation of the JT black hole. Remarkably, our result exactly reproduced an analogue of the Page curve for the entanglement negativity obtained through the random matrix technique derived in [70]. In the second scenario, we varied the length of only one of the two intervals while keeping the $\alpha$ parameter and the sum total of the length of the intervals fixed. Quite interestingly, once again the behaviour of entanglement negativity determined from our computation precisely matched with an analogue of the corresponding Page curve obtained from the random matrix technique in [70]. In the third case, we fixed the $\alpha$ parameter and the length of one of the subsystems, and determined the entanglement negativity of the adjacent intervals by varying the length of the other interval. We described the behaviour of entanglement negativity obtained in all the three cases from the dynamics of the entanglement between the Hawking quanta collected in the bath subsystems involved and their corresponding islands.

Subsequently, we considered the mixed state configuration of two disjoint intervals in a

bath coupled to an evaporating JT black hole. We described this case by further dividing it into two scenarios. In the first case, we determined the entanglement negativity by varying the length of one of the disjoint intervals in question, while keeping the length of the other interval and the distance between them fixed. In the second case, we once again kept the length of one of the interval fixed but varied the length of the other interval and the distance between them. In both of the above mentioned scenarios we could once again explain the behaviour of entanglement negativity through the interplay of the entanglement between the subsystems, and the purification due to the appearance of islands in the two dimensional effective theory. Finally, we considered the mixed state configurations involving the JT black hole and a part of the Hawking radiation/bath. In the model developed in [20] the quantum mechanical degrees of freedom dual to the evaporating JT black hole are described by an interval $[0, b]$ in the limit $b \rightarrow 0$. We computed the entanglement negativity between such an interval describing the quantum mechanical degrees of freedom and another interval which is completely inside the bath subsystem, both for adjacent and disjoint configurations. As earlier we could interpret all the results we obtained from the island construction in two dimensional effective theory.

Furthermore in the appendix, we have obtained the entanglement negativity of various mixed state configurations in a bath coupled to a two dimensional extremal JT black hole obtained from a partial dimensional reduction of the pure $AdS_3$ space time as described in [20]. The mixed state configurations we considered here included the adjacent and the disjoint intervals involving bath and black hole. Note that as is well known the extremal black holes are at a vanishing Hawking temperature, and therefore they do not evaporate. Hence, their exact role in the information loss paradox is unclear and a definite interpretation of the results for this case remains elusive due to the subtleties involved.

Our results lead to several possible future directions. It would be very exciting to compare the entanglement negativity we computed for various configurations to the corresponding results obtained in other models involving the double holographic formulations examined in [13–15, 17, 18, 24, 25]. Investigation of other mixed state entanglement and correlation measures such as reflected entropy, entanglement of purification and odd entropy for bipartite systems in the present model developed in [20] should be significant in further understanding the correlation structure of the Hawking radiation. Exploring the behaviour of the entanglement negativity in other constructions involving the geometrization of the black hole evaporation process and the island formation considered in [10, 11, 22] might lead to novel insights about how the Hawking radiation encodes the information about the black hole interior. It would also be quite interesting to examine the behaviour of the entanglement negativity when gravitating baths are involved [16,21,23,88]. We leave these fascinating issues for future investigations.

# 5 Acknowledgement

We are grateful to Ashish Chandra and Mir Afrasiar for useful discussions.

# A Extremal JT black holes

In this appendix we determine the entanglement negativity for various mixed state configurations in a bath coupled to an extremal black hole in JT gravity obtained through the partial dimension reduction of the pure $AdS_3$ space time as will be reviewed below. Unlike the non-extremal JT black hole, the extremal black hole has a vanishing Hawking temperature and hence in this case it does not evaporate. As a result, the explicit role of the extremal black hole

in the information loss paradox remains unclear. Also note that in the model described in [20] the $\alpha = \frac{\Phi_r}{2\pi}$ parameter does not control the time dependence anymore. Therefore, a coherent interpretation of the results we obtain below remains as an open issue for future investigation.

## A.1 Review of the partial dimensional reduction

In this section we briefly review how extremal black hole in JT gravity can be obtained through partial dimension reduction from a three dimensional pure $AdS_3$ [20]. The metric for the pure $AdS_3$ space time in Poincaré coordinates is given as

$$ds^2 = \frac{L_3^2}{z^2}(-dt^2 + dz^2 + dx^2), \tag{42}$$

where $L_3$ denotes the $AdS_3$ length scale. On using the coordinate transformation $z = \frac{L_3^2}{r}$ and $x = L_3 \varphi$, the above equation may be written as

$$ds^2 = -\frac{r^2}{L_3^2}dt^2 + \frac{L_3^2}{r^2}dr^2 + r^2 d\varphi^2, \tag{43}$$

where $\varphi$ corresponds to the angular coordinate with a period $2\pi$. Once again the above metric is of the form eq. (7) which leads to an extremal black hole in JT gravity upon a partial dimension reduction of pure $AdS_3$ in Poincaré coordinates as described in [20]

$$ds^2 = \frac{-4L_3^2\, dX^+ dX^-}{(X^+ - X^-)^2} + \frac{4L_3^4\, d\varphi^2}{(X^+ - X^-)^2}, \tag{44}$$

where $X^\pm = t \pm z$ are the light cone coordinates. The authors made the following identification in order to identify the above metric with $AdS_2$ metric in Poincaré coordinates

$$\Phi = \Phi_0 + \Phi_r \frac{\sqrt{g_{\varphi\varphi}}}{\ell^2} = \Phi_0 + \frac{2\Phi_r}{X^+ - X^-}, \tag{45}$$

where $\Phi$ corresponds to the dilaton in two dimensional JT gravity. As described in [20], upon partial dimensional reduction in the $\varphi$ direction (i.e integrating over some angle $2\pi\alpha$, where $\alpha \in (0,1]$), the spacetime consists of two parts, namely the extremal JT black hole and the rest corresponds to a two dimensional bath subsystem described by a CFT.

Consider the subsystem $[0, b]$ in the CFT/bath region, which also includes the quantum-mechanical degrees of freedom. The entanglement entropy of this subsystem may be obtained by computing the length of a geodesic in the three dimensional bulk. In this case, the length of a geodesic homologous to a boundary subsystem of length $\Delta\varphi$ in coordinates $(t, r, \varphi)$ is given by

$$\mathcal{L}_{\Delta\varphi} = 2L_3 \log \Delta\varphi + \text{UV cutoff}. \tag{46}$$

Hence, the subsystem described by an interval i.e $[0, b]$ in the bath, the entanglement entropy of the boundary subsystem is then obtained using eq. (46) and the RT formula as [20]

$$S = \frac{1}{4G}\left(\Phi_0 + 2\log\frac{\Phi_r + 2b}{L}\right), \tag{47}$$

where we have made the following identifications $L_3 = L$, $G \equiv G^{(2)} = \frac{G^{(3)}}{L}$, $\Phi_r = 2\pi L\alpha$ and the UV cut off is absorbed in the background value of the dilaton denoted as $\Phi_0$.

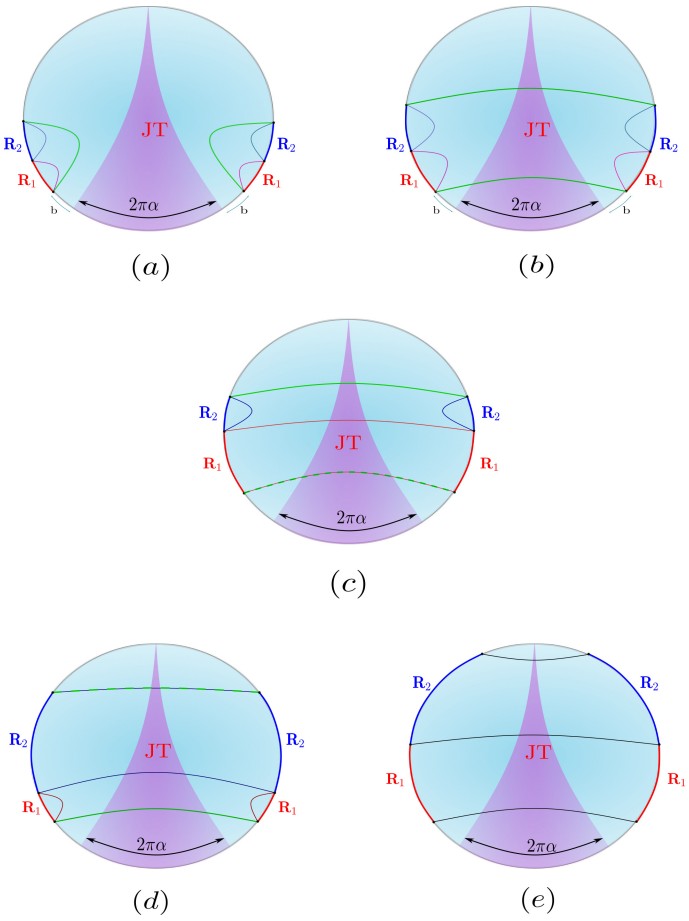

Figure 17: Schematic of different possible phases for adjacent intervals configuration in a bath.

## A.2 Entanglement Negativity

To begin with we determine the Renyi entropy of order half for a generic boundary subsystem $A$ described by an interval of length $\Delta\varphi$ in the bath, which we require to compute the entanglement negativity of various configurations in question. The Renyi entropy of order half for such an interval $A$ can be obtained using eqs. (19), (18) and (46) as follows

$$S^{(1/2)}(A) = \Phi_0 + \frac{c}{2}\log\Delta\varphi\,, \tag{48}$$

where we have used the Brown-Henneaux formula $c = \frac{3L}{2G_N^{(3)}}$ [74] and absorbed the UV cutoff in $\Phi_0$. We will use the above expression to compute the entanglement negativity in the following subsections.

### A.2.1 Adjacent Intervals in Bath

In this subsection we consider the mixed state configuration of two adjacent intervals $R_1 \equiv [b, b + \ell_1]$ and $R_2 \equiv [b + \ell_1, b + \ell_1 + \ell_2]$ in the bath as shown in fig. 17. We observe that depending on the size of the intervals $R_1, R_2$, we have different phases which we describe below.

**Phase I**

In phase I, the sizes of both the intervals $R_1$ and $R_2$ are small such that the corresponding RT surfaces are disconnected. This is depicted in fig. 17(a). The entanglement negativity for the adjacent intervals in this phase may be obtained using eqs. (48) and (20) as

$$\mathcal{E}(R_1 : R_2) = \Phi_0 + \frac{c}{2} \log\left[\frac{\ell_1 \ell_2}{L(\ell_1 + \ell_2)}\right].\tag{49}$$

**Phase II**

In this phase, the RT surfaces corresponding to $R_1$ and $R_2$ are disconnected but connected for $R_1 \cup R_2$. This is shown in fig. 17(b). The entanglement negativity for the configuration of adjacent intervals for phase II may now be computed using eqs. (48) and (20) as

$$\mathcal{E}(R_1 : R_2) = \Phi_0 + \frac{c}{4} \log\left[\frac{\ell_1^2 \ell_2^2}{L^2(\Phi_r + 2b)(\Phi_r + 2b + 2\ell_1 + 2\ell_2)}\right],\tag{50}$$

where $\Phi_r = 2\pi L \alpha$.

**Phase III**

The phase III is described by the configuration in the subsystem $R_1$ is large enough to have connected RT surfaces whereas the RT surfaces corresponding to $R_2$ are disconnected as depicted in fig. 17(c). We now employ eqs. (48) and (20) to obtain the entanglement negativity for this phase as

$$\mathcal{E}(R_1 : R_2) = \Phi_0 + \frac{c}{4} \log\left[\frac{(\Phi_r + 2b + 2\ell_1)\ell_2^2}{L^2(\Phi_r + 2b + 2\ell_1 + 2\ell_2)}\right].\tag{51}$$

**Phase IV**

In this phase the RT surfaces corresponding $R_1$ are disconnected whereas the same for $R_2$ are connected. It is shown in fig. 17(d). The entanglement negativity for adjacent intervals in phase IV may now be computed using eqs. (48) and (20) as follows

$$\mathcal{E}(R_1 : R_2) = \Phi_0 + \frac{c}{4} \log\left[\frac{(\Phi_r + 2b + 2\ell_1)\ell_1^2}{L^2(\Phi_r + 2b)}\right].\tag{52}$$

**Phase V**

In this phase, the RT surfaces corresponding to both the intervals $R_1$ and $R_2$ are connected as depicted in fig. 17(e). The entanglement negativity for the configuration of the adjacent intervals for phase V may now be given as

$$\mathcal{E}(R_1 : R_2) = \Phi_0 + \frac{c}{2} \log\left[\frac{\Phi_r + 2b + 2\ell_1}{L}\right].\tag{53}$$

### A.2.2 Disjoint Intervals in Bath

We consider the mixed state configuration of two disjoint intervals in the bath described by $R_1 \equiv [b, b + \ell_1]$ having length $\ell_1$ and $R_2 \equiv [b + \ell_1 + \ell_s, b + \ell_1 + \ell_s + \ell_2]$ of length $\ell_2$. These two intervals are separated by $R_s \equiv [b + \ell_1, b + \ell_1 + \ell_s]$ having length $l_s$. We see that depending

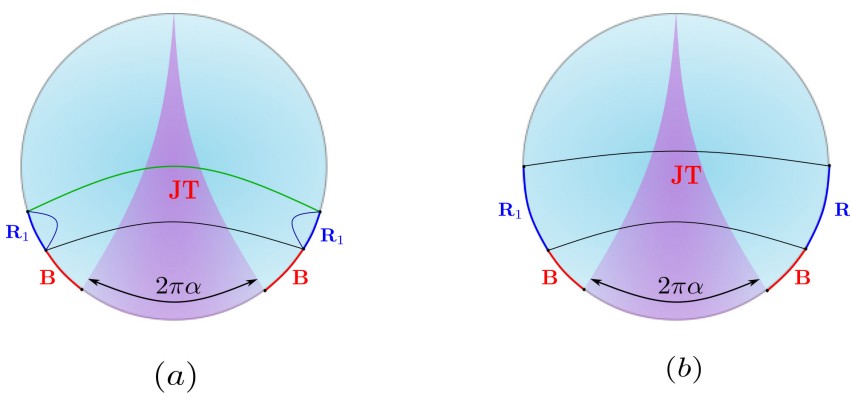

Figure 18: Figure (a) depict the RT surface for $R_1$ in the disconnected phase where as the schematic in figure (b) corresponds to the connected phase of the RT surface.

on the size of the intervals $R_1$, $R_2$, there are many possible phases some of which we discuss below.

**Phase I**

In this phase, we take the size of the subsystems $R_1$ and $R_2$ such that the RT surfaces corresponding to the subsystem $R_1 \cup R_s$ are connected but disconnected for $R_2 \cup R_s$. We also consider $R_s$ to be small so that its RT surfaces are always disconnected. The entanglement negativity for the configuration of disjoint intervals in bath may now be obtained using eqs. (48) and (22) as

$$\mathcal{E}(R_1 : R_2) = \frac{c}{4} \log \left[ \frac{(\ell_2 + \ell_s)^2 (\Phi_r + 2b + 2\ell_1 + 2\ell_s)}{\ell_s^2 (\Phi_r + 2b + 2\ell_1 + 2\ell_s + 2\ell_2)} \right]. \tag{54}$$

**Phase II**

This phase is described by the configuration in which $R_1$ and $R_2$ are large enough such that the RT surfaces for $R_1 \cup R_s$ and $R_2 \cup R_s$ are connected. We now employ eqs. (48) and (22) to compute the entanglement negativity for this phase as follows

$$\mathcal{E}(R_1 : R_2) = \frac{c}{4} \log \left[ \frac{(\Phi_r + 2b + 2\ell_1)(\Phi_r + 2b + 2\ell_1 + 2\ell_s)}{\ell_s^2} \right]. \tag{55}$$

The entanglement negativity for other remaining phases can be obtained in a similar manner described above.

### A.2.3 Adjacent Intervals involving Black hole and Bath

We consider the mixed state configuration described by the adjacent intervals $B \equiv [0, b]$ (which also includes the QM degrees of freedom) of length $b$ and $R_1 \equiv [b, b+\ell_1]$ of length $\ell_1$ as shown in fig. 18. We see that there are two phases for this configuration depending on the size of interval $R_1$ which are discussed below.

**Phase I**

For the configuration of the adjacent intervals in phase I, the RT surfaces corresponding to the subsystem $R_1$ are disconnected as depicted in fig. 18(a). The entanglement negativity for the

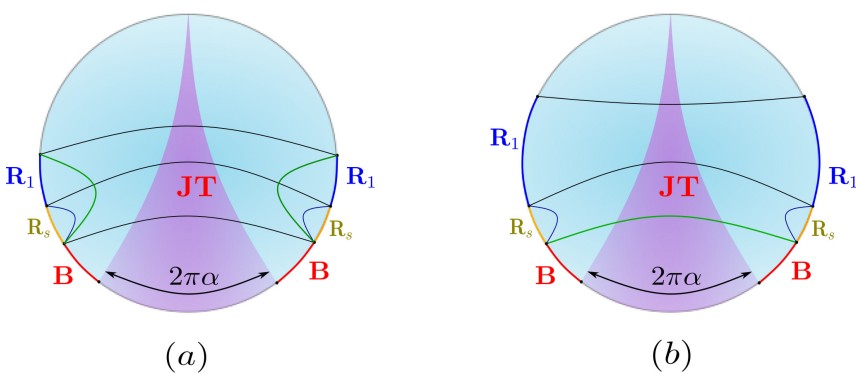

Figure 19: Schematic of different phases for configuration of disjoint intervals.

adjacent intervals in this phase may now be obtained using eqs. (48) and (20) as

$$\mathcal{E}(B:R_1) = \Phi_0 + \frac{c}{4} \log\left[ \frac{(\Phi_r + 2b)\ell_1^2}{L^2(\Phi_r + 2b + 2\ell_1)} \right]. \tag{56}$$

**Phase II**

In phase II, we have connected RT surfaces for the interval $R_1$ which is shown in fig. 18(b). The entanglement negativity for the adjacent intervals in phase II may be obtained using eqs. (48) and (20) as follows

$$\mathcal{E}(B:R_1) = \Phi_0 + \frac{c}{2} \log\left[ \frac{\Phi_r + 2b}{L} \right]. \tag{57}$$

From the above expressions it is clear that the entanglement negativity increases as we increase the size of the subsystem $R_1$ and become constant after a certain value of $\ell_1$.

### A.2.4 Disjoint Intervals involving Black hole and Bath

For the mixed state configuration of two disjoint intervals, we consider the subsystems $B \equiv [0, b]$ of length $b$ (which also includes the QM degrees of freedom), and $R_1 \equiv [b + \ell_s, b + \ell_s + \ell_1]$ having length $\ell_1$, which are separated by $R_s \equiv [b, b + \ell_s]$ of the length $\ell_s$. This configuration is depicted in fig. 19. We observe that depending on the size of interval $R_1$, we have different phases which are described below.

**Phase I**

The RT surfaces for the subsystem $R_1 \cup R_s$ in this phase are disconnected as shown in fig. 19(a). Note that we consider $R_s$ to be small such that RT surfaces corresponding to it are always disconnected. We now employ eqs. (48) and (22) to obtain the entanglement negativity for this phase as

$$\mathcal{E}(B:R_1) = \frac{c}{4} \log\left[ \frac{(\ell_1 + \ell_s)^2(\Phi_r + 2b + 2\ell_s)}{\ell_s^2(\Phi_r + 2b + 2\ell_1 + 2\ell_s)} \right]. \tag{58}$$

**Phase II**

In this phase, the RT surfaces for $R_1 \cup R_s$ are connected as depicted in fig. 19(b). As earlier we find the entanglement negativity for phase II, utilizing eqs. (48) and (22) as follows

$$\mathcal{E}(B:R_1) = \frac{c}{4} \log\left[ \frac{(\Phi_r + 2b)(\Phi_r + 2b + 2\ell_s)}{\ell_s^2} \right]. \tag{59}$$

Once again from the above expressions, we note that the entanglement negativity increases as a function of the size of the subsystem $R_1$ first and eventually saturates.

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
