# Peer review of "Page Curve for Entanglement Negativity through Geometric Evaporation"

_SciPost Physics, doi:SciPost Phys. 12, 004 (2022)_

## Round 1 · Referee Report · Anonymous · 2021-8-22

Weaknesses
1- Some notations and concepts are not explained properly.
2-Do not provide enough analytical results.
3-Too many similar numerical plots but the conclusions and lessons from these plots are not clear.
4-The role of the dynamics of the evaporating BH in the entanglement negativity is not explained.
Report
This manuscript studied the entanglement negativity of various subsystems in 2D JT gravity coupled to a bath. Particularly, the 2D model is proposed in Ref[20] and derived from the 3D BTZ black hole by performing a partial dimension reduction of the $ \phi$-direction. By making the reduction angle $\alpha$ time-dependent, it realizes the 2D black hole evaporation in a geometric way.
In this manuscript, the authors numerically studied the holographic entanglement negativity of adjacent intervals and also disjoint intervals. However, I think the authors should be able to provide more analytical results. So I would like to request the authors to add more calculations and resubmit the manuscript. My two main requests are
01- Add analytical results for the entanglement negativity of different systems. For a single interval, the Renyi entropy of order half has been shown in eq.(29).
The full results for $\mathcal{E}$ are expected to be complicated. However, most plots in this manuscript present very simple behaviors. Based on eqs.(29,30), the authors can explain the linear growth and derive the growth rate. And in some regimes, the authors should use their formula to illustrate the appearance of the plateaus.
02-In eq(33), the authors present the entanglement negativity with islands. Although the holography entanglement negativity is enough to derive the results, it is still worth exploring the island formula in detail. Of course, it is not easy to calculate the contributions $\mathcal{E}^{\mathrm{eff}}$. But the leading contribution is given by the area term. In this geometric evaporating BH model, the time evolution of the dilaton is solvable. This makes it possible to match the holographic result by using the island formula.
I have some other minor questions or suggestions as listed in "Requested Changes".
Requested changes
1- In section 2.1, the authors should add the expression for the time-dependent $\alpha(t)$ and explain why this can result in an evaporating black hole.
2- For the results shown in this manuscript, I do not understand what is the role of the evaporating BH. For example, most figures (eg, 6,9, 11,12,14,16) for the entanglement negativity correspond to a fixed value of $\alpha$. What is the influence of BH evaporation on the entanglement negativity of different subsystems?
3- The interval B should be labeled in the figures.
4- In equation (29), the authors introduce a parameter $\phi'$ as the distance between subsystem R and interval B. But I do not understand what is the meaning of this extra parameter. Is this used to regularize the limit $b \to 0$? But this is can be done by putting a cut-off as $b \to \phi'$. From eq(29), you can redefine $b \to b+\phi'$ to absorb that parameter $\phi'$.
5- For all the numerical plots with various parameters, I guess the authors fix the AdS radius as 1. I prefer to choose dimensionless parameters like $L/\beta, b/L$ in numerics.
6- In most captions of figures, the authors choose $c=500$. But this notation $c$ and its connection to $G_N$ is not introduced in the context and $c$ never appears in any formulas.
7- In figure 2, 3, the x-axis is $1-\alpha$. Do the authors want to use this as a time direction? From the evaporation model discussed in ref[20], it is related to the tilded time. See their eq(4.17)
\[\alpha(\tilde{t}) = 1- \frac{A}{2}\tilde{t}.\]
If this is the model the authors want to discuss, this point should be explained in the context.
8- Figure 5 is named as the Page curve for entanglement negativity. But I do not understand the meaning of this because it only represents a fixed time slice (a fixed $\alpha$).
9- According to the context after eq.(29), the subsystem R is parametrized by the angle $\phi_i$. But in the eq. (29), the angle $|\phi_2-\phi_1|$ represents a length like $b, L$. So the eq.(29) may have some typos or the explanation of $\phi_i$ is not correct. And I think it is not a good idea to use the angles and length $b, l_1, l_2$ in the meantime.
10-In eq.(33), the authors should explain what is the meaning of area $\mathcal{A}^{1/2}$ in 2D JT gravity and its connection to the value of dilaton because the surface $Q$ is just a point in this model.
Author: Vinay Malvimat on 2021-10-03 [id 1798]
(in reply to Report 1 on 2021-08-22)
We would like to thank the referee for the interesting questions raised. Our detailed response to the referee's comments and the resolution of these issues is described below. We have appropriately made the requested changes to our manuscript as suggested by the referee.
Below is our response to the two main requests by the referee and the list of the corresponding changes made.
- As described in the line above eq.(4.30) in ref [20], the simple linear behaviour of the entanglement entropy is valid in the approximation $\beta<<\Phi_r^0$ ( $\Phi_r^0=2\pi L$ where $L$ is the AdS radius which we have set to unity $L=1$ ). This is also true for the holographic entanglement negativity as it is given by a linear combination of the Renyi entropies of order half. Hence, all our plots are also valid within the approximation $\beta<<\Phi_r^0$. As suggested by the referee, we have provided the analytic results for the entanglement negativity for various configurations which display simple behaviours in the above described approximation. We have also added the above discussion in the beginning of section 3 in page 9 of our revised manuscript.
- This is an interesting direction suggested by the referee. The island construction in the lower dimensional effective theory equivalent to a geometric evaporating black hole in ref [20] involves two JT branes on either side of a Minkowski bath coupled to them. Furthermore, the one dimensional equivalent model involves a $CFT_2$ with quantum mechanical degrees of freedom at its two boundaries. These do not conform to the standard effective theories which have been examined in the existing literature on island constructions. In ref [20] the authors have not provided a complete picture of the equivalent 2D or 1D effective theories and the corresponding island construction for entanglement entropy in the lower dimensional effective model from first principles. Hence, at this stage the corresponding equivalent model for islands is not understood in its full technicality . Currently we can only interpret the results we have obtained through the partial dimensional reduction from the appearance and disappearance of the corresponding islands for the entanglement negativity of various cases considered in our article. For the above reasons suggested computation needs to wait for the complete understanding of the island construction in the lower dimensional effective theory in its full technicality.
Below is the list of changes made as a response to the minor questions raised by the referee.
- Making $\alpha$ time dependent corresponds to a time-varying (renormalized) dilaton obtained via the partial dimensional reduction of a BTZ black hole. The linearly decreasing profile of $\alpha$ leads to an outgoing energy flux of the matter CFT$_2$ present in the heat bath which is dual to the rest of the BTZ black hole spacetime remaining after the partial dimensional reduction.. As described in ref [20], this outgoing energy flux simulates a black hole evaporation process. We have discussed the above in section 2.1 of our revised manuscript and have added the expression for time -dependent $\alpha(t)$ in eq. (12).
- We agree that many of the plots and computations we demonstrate in our article are for fixed $\alpha$. However we would like to emphasize that our computations are analogous to Page's original article for entanglement entropy given in ref [81] of our revised manuscript. A similar computational setting was utilized to derive a Page curve for the entanglement negativity through random matrix theory in ref [70] of our revised manuscript. The authors in both of the above mentioned references did not examine the time dependent behaviour of the entanglement entropy or negativity but studied these measures by varying the subsystem size. In fact the $\alpha$ parameter which controls the extent to which the black hole evaporates in ref [20] also describes an angle. Hence, it is valid to examine the entanglement negativity as a function of the size of the subsystems for a fixed $\alpha$ or vary the $\alpha$ parameter by keeping some function of the size of the subsystems fixed. However in order to compare our results with corresponding results in the random matrix theory we kept the parameter $\alpha$ fixed in some of our computations. A brief discussion of these issues may be found in the beginning of section 3 in our revised manuscript. We would like to emphasize that we had examined the influence of the black hole evaporation on the entanglement negativity of two adjacent intervals by varying $\alpha$ and its plot is depicted in figure 4. The behaviour of the entanglement negativity for this case also exactly matched with the corresponding results from random matrix theory obtained in ref [70] of our revised manuscript.
- We have shown the interval $B=[0,b]$ describing the quantum mechanical degrees of freedom dual to the JT black hole in all the figures in the revised version of our manuscript as suggested by the referee except in figure 7 and 8 which are only utilized for illustrating entanglement negativity islands, to avoid notational clutter .
- We would like to emphasize that $\phi'$ cannot be absorbed in $b$. The reason for this is as follows. The interval $[0,b]$ corresponds to the quantum mechanical degrees of freedom dual to the 2d JT black hole only in the limit $b\to 0$ as described in ref [20] of our revised manuscript. Hence, the length of this interval given by $b$ has to be small. However $\phi'$ is introduced to keep the subsystem in the radiation/bath to be generic as it need not always be adjacent to the interval $[0,b]$. Hence, all our results hold even if one sets $\phi'=0$. We have added a footnote in page 11 of our manuscript clarifying this point.
- We agree that we missed specifying the fact that we have set the AdS length scale to unity, $L=1$ for our calculations. We have mentioned this in our revised manuscript wherever necessary.
- $c$ is the central charge of the matter CFT$_2$ describing the holographic dual of the portion of BTZ spacetime which is not dimensionally reduced. The central charge of the dual CFT$_2$ is related to $G_N$ through the well known Brown-Henneaux relation $c=\frac{3L}{2 G_N}$. We have added this relation below eq.(12) in section 2.1 of our revised manuscript.
- We would like to thank the referee for pointing out this significant issue, which had not been properly described in our earlier submission. We indeed wish to use $1-\alpha$ as a (rescaled) time variable and this is now addressed in eq. (12) of our revised manuscript. We would like to further emphasize the fact that we are working in what are described as the tilded coordinates (describing linear evaporation) in ref [20] and have chosen to omit the tildes for brevity. This is clarified in footnote 1 on page 7 of our revised manuscript.
- This is very similar to point 2 above. In figure 5 we examined the behaviour of the entanglement negativity between two subsystems with respect to their relative size. We call figure 5 as the Page curve in order for it to be consistent with the corresponding plot determined in ref [70] of our revised manuscript where the authors investigated the same as a function of the size of their Hilbert spaces. However, we would like to emphasize that we have examined the entanglement negativity of two adjacent subsystems by varying $\alpha$ whose plot is depicted in figure 4. The behaviour of the entanglement negativity we obtained in figure 4 exactly matched with the corresponding results from random matrix theory obtained in ref [70] of our revised manuscript .
- We admit that there is a notational clutter here. In our revised manuscript the subsystem $R$ is parametrized by the angular positions $\varphi_1$ and $\varphi_2$ of its endpoints such that it has an angular extension of $\Delta\varphi\equiv |\varphi_1-\varphi_2|$. The lengths of the backreacting branes are now parametrized by the lengths $\ell_{\varphi_1}=L\varphi_1$, $\ell_{\varphi_2}=L\varphi_2$ and $\ell_{\Delta\varphi}=L\Delta\varphi$ as described in eqs. (23) through (26) of our revised manuscript. In eq. (30) we have described the Renyi entropies in terms of the actual length of the subsystem $R$ as $\ell_{\Delta\varphi}=L\Delta \varphi$. Hence, in the revised version of our manuscript, the expressions for the entanglement negativity of all the cases considered are described in terms of the lengths of the various subsystems involved.
- In 2d JT gravity, the value of the dilaton at some point is interpreted as the area from the higher dimensional perspective. In this context, if one considers the replica geometry corresponding to the $n$-th Renyi entropy for a subsystem which is just a point at the boundary of the $(1+1)$-dimensional CFT, the natural generalization of the quantum extremal surface is a backreacting cosmic brane of finite tension. The Renyi entropy is subsequently obtained in terms of the area of the cosmic brane which due to backreaction of the replica geometry depends on the replica parameter. In the framework of JT gravity, the analogue of this backreacting cosmic brane is a $n$-dependent dilaton field $\phi^{(n)}$ picking up contributions from the conical singularities present in the replica manifold. Therefore, the order half area term in eq. (33) corresponds to the analytic continuation of the dilaton $\phi^{(n)}$ to $n\to\frac{1}{2}.$ We have added a footnote on page 15 of our revised manuscript describing this subtlety.
Author: Vinay Malvimat on 2021-10-03 [id 1800]
(in reply to Report 2 on 2021-09-04)We would like to thank the referee for the interesting questions raised. Our detailed response to the referee's comments and the resolution of these issues is described below. We have appropriately made the requested changes to our manuscript as suggested by the referee.
Below is our response and the list of changes made.
The linear characteristics of the Page curve for entanglement entropy obtained through the model in ref. [20] is valid only for $\beta<<\Phi_r^0$ ( $\Phi_r^0=2\pi L$ where $L$ is the AdS radius which we have set to unity $L=1$ ). This is clearly expressed in the line above eq.(4.30) in ref. [20]. Away from this approximation, that is for larger values of $\beta$, the Page curve for entanglement entropy obtained through the partial dimensional reduction model of ref. [20] itself deviates from its linear behaviour. This is true for the holographic entanglement negativity as in our construction it is given by a linear combination of the Renyi entropies of order half. Hence all our plots are also valid within the approximation $\beta<<\Phi_r^0$. Away from this approximation, the plots for entanglement negativity differ from those shown in our article. We have added this discussion in the beginning of section 3 in page 9 of our revised manuscript.

---

## Round 1 · Referee Report · Anonymous · 2021-9-4

Strengths
The connection between black hole and random matrix theory.
Weaknesses
- The relation between the area of back-reacting cosmic branes in AdS$_3$ and geodesic lengths is valid only for spherical entangling surfaces.
- The manuscript lacks of clarity in some part.
Report
In this work, the authors compute holographic entanglement negativity and analyse its dependence in terms of the relevant scales in the system (referred to as analogues of Page curve for negativity) for pure and mixed states in a bath coupled to a two-dimensional Jackiw-Teitelboim (JT) black hole.
To be more specific, the system is holographically engineered via a ``dimensional reduction'' proposed by Verheijden et al. (ref. [20] in the manuscript), where a JT black hole is obtained after a partial dimensional reduction of a three-dimensional BTZ black hole. The rest of the three-dimensional geometry is dual to a conformal field theory (CFT$_2$).
The crucial points in ref [20] for this manuscript are a) the dynamical evaporation of a JT black hole is realised by making the parameter controlling the reduction time-dependent, and b) dynamical quantities like entanglement of radiation can be simply computed from a three-dimensional geometrical point of view, as shown already in ref. [20].
Entanglement negativity (or logarithmic negativity) is an entanglement measure suitable for mixed states, it has the benefit to detect only quantum correlations and to be computable in quantum field theory settings.
Here, holographic entanglement negativity (HEN) is computed according to a series of conjectures originally proposed in ref. [40, 43, 45] in the manuscript.
In essence, the authors expressed the HEN in terms of the dual of Renyi entropies of order $1/2$, which are given by the minimum of geodesic lengths computed according to the dictionary proposed in ref. [20].
Equipped with this, the authors investigate the analogues of a Page curve for mixed states.
Their results qualitatively match the findings of Shapourian et al. (ref. [54] in the manuscript) for random mixed states.
There are various scales to be tuned here (e.g. the relative size between the subsystems $A_i$, the relative size between the whole subsystem $A$ and the bath $B$, the distance between the subsystems $A_i$ in the disjoint case). The resulting phase diagram of negativity as a function of these scales is rather reach and can distinguish between different phases, depending on the parameters which are kept fixed or not. The authors describe the different cases in detail.
The problem faced in the manuscript and the methods used to compute entanglement negativity are not new, however the authors are able to add a novel and interesting result to this very active line of research by combining previous techniques.
Holographic entanglement negativity in the bath with a coupled JT black holes was already discussed in a previous paper by Kumar Basak et al. (ref [48] in the manuscript). Here the novelty is the use of the setting proposed in ref.[20] and the systematic investigation of an analogue of a Page curve for the negativity.
In my opinion the work contains some interesting material which deserves to be published.
Before publication, the authors should address the following points.
Requested changes
1. Introduction. Page 1 and largely page 2 are devoted to review previous results. However these pages contain information which are not directly used in the actual analysis. In my opinion, the reader would benefit from a more streamed and concise introduction. Moreover, the novel aspects on the manuscript are summarised in 5 lines on page 4. This section is "unbalanced' and misguide the reader. I understand this is a general comment, but I strongly advice to improve the narrative of the introduction, for example I do not see the reason to describe so extensively results of ref. [48], when their techniques and results are not directly used in this work.
2. Section 2.3, page 8. Below eq. (19) the authors are referring to the wrong paper, I believe ref [45] (and not [43]) should be mentioned.
3. Section 2.3. The authors list eq. (19) and (21) which are the expressions used later to computed the negativity. For reader's convenience it would be useful to report here also the analogue expression for adjacent intervals.
4. Page 10, eq. (28) shows the minimum of four configurations and not three as said above eq. (28).
5. Overall, on the plots not all the numerical values are indicated, for example $L$ is missing.
6. All the plots are (consistently) obtained for the same values of certain parameters, such as $\Phi_0$ and $\beta$. Did the authors try other values of temperature and cutoff? If so, is there any qualitative difference?
7. I would like to have some clarifications to fig. 6. This seems essentially a four-partite system, four lengths are in play (though not all independent) and it is not directly analogue to the configurations examined in the random matrix theory as in ref. [53]. Could the authors comment on this?
8. Figures 11 and 6 are very similar, even though obtained for disjoint and adjacent intervals respectively. Usually in field theories logarithmic negativity for adjacent and disjoint intervals has different behaviours. Can the authors comment on this result?
9. Sections 3.1.4, and 3.1.5. Also, it seems that there is no qualitatively different behaviour when the JT black hole is involved, for adjacent and disjoint intervals. But how solid is this result? The authors are using the same numerical value for $b$ as in previous sections. Can the authors comment on this?
10. HEN is obtained as a certain linear combination of geodesic lengths, which, for spherical entangling surfaces, are proportional to back-reacting cosmic branes in AdS$_3$, which in turn are conjectured to be dual to Renyi entropies of order $1/2$.
The derivation of the expressions for the holographic entanglement negativity is somewhat confusing. It starts with the expression in terms of geodesic lengths, according to the proposal ref [40]. On the other hand the geodesic lengths are proportional to the area of back-reacting cosmic branes in AdS$_3$ (at least for certain symmetric configurations of the intervals), and these are related to Renyi entropies of order 1/2. Again these are expressed in terms of lengths computed according to the prescriptions of ref. [20]. Why do the authors cannot formulate directly the initial expression for example in eq. (14) in terms of eq. (3, 4) (assuming that we are taking the min)? It would be useful to see a clarification on this point.
11. Finally, there are various typos that should be corrected (punctuations and spaces, page $\to$ Page, algbraic $\to$ algebraic etc. etc.).

---

## Round 2 · Referee Report · Anonymous · 2021-10-7

Report

The authors have addressed the issues I pointed out, and I am content with the corrections and the various improvements made in the revised version. In my opinion the manuscript can be published.

---

## Round 2 · Referee Report · Anonymous · 2021-10-12

Report

The authors have addressed most issues raised in the last report. Therefore, I would recommend the publication of the paper.

---

## Round 2 · Author Response

We have examined the reports of the two referees for our submission scipost\_202107\_00042 entitled ``Page Curve for Entanglement Negativity through Geometric Evaporation". We would like to thank the referees for the interesting questions raised. We have provided our detailed response to the referee's comments and the list of changes made in the authors' response section.

---

## Round 2 · List of Changes

We have provided the complete list of changes made in the authors' response section.

---

## Editorial Decision

published